# What's in the Bottle?
# A Survey and Roadmap of Concept Bottleneck Models

**Patrick Knab**[1*]    **David Steinmann**[2,3*]    **Christian Bartelt**[1]    **Kristian Kersting**[2,3,4]
**Bernt Schiele**[5,6]    **Thomas Seidl**[7,8]    **Udo Schlegel**[7,8†]    **Wolfgang Stammer**[5,6†]

[1] *Clausthal University of Technology*    [2] *TU Darmstadt*    [3] *hessian.AI*    [4] *DFKI*
[5] *Max Planck Institute for Informatics, SIC*    [6] *RTG Neuroexplicit Models*
[7] *LMU München*    [8] *Munich Center for Machine Learning*

**Reviewed on OpenReview:** *https://openreview.net/forum?id=IF5vnqxBEW*

## Abstract

Concept Bottleneck Models (CBMs) are interpretable learning architectures that factor predictions through intermediate, ideally human-understandable concepts, enabling explicit and inspectable reasoning. Although CBM research has gained substantial momentum in recent years, this growth has also revealed numerous open challenges and a fragmented set of methodological choices. In this work, we systematically review the CBM literature, identify previously unidentified core components and challenges, and propose a unified taxonomy. Based on this taxonomy, we provide a detailed categorization of existing works. We hereby discuss current challenges for the CBM paradigm and outline important directions to extend it beyond its current scope. Overall, this survey aims to consolidate the CBM landscape, clarify open issues, and provide guidance for developing future models.

## 1 Introduction

Artificial intelligence (AI) models demonstrate exceptional performance in a wide range of applications, but this success often comes at the cost of interpretability. In high-stakes domains such as healthcare, finance, and law, this lack of transparency and control is particularly problematic (Poeta et al., 2023; Von Eschenbach, 2021; Knab et al., 2025a). While fully *interpretable-by-design* models (Rudin et al., 2022; Gupta & Narayanan, 2024; Böhle et al., 2024; Lee et al., 2024) allow complete inspection of their reasoning process, they commonly exhibit a reduced predictive accuracy, highlighting a persistent accuracy–interpretability trade-off (Espinosa Zarlenga et al., 2022). Moreover, achieving full interpretability for complex input data such as images is often impractical.

A recent direction has begun to address this challenge through Concept Bottleneck Models (CBMs) (Koh et al., 2020; Lage & Doshi-Velez, 2020; Stammer et al., 2021; Losch et al., 2021), which ground predictions in human-understandable concepts. CBMs explicitly decompose predictions into two stages: first, predicting a set of human-interpretable concepts, then using only these concepts to produce the final prediction of the model (cf. Fig. 1a). This bottleneck structure enforces selective interpretability, i.e., the model's reasoning becomes transparent at the concept level, even when the underlying feature extraction remains complex. Importantly, initial work demonstrated that given high-quality concepts, even simple and explainable linear predictors can achieve strong performance while enabling desirable properties such as interpretable feature interactions, targeted interventions on mispredicted concepts, and modular reasoning.

At the same time, CBMs introduce novel challenges specific to their architecture, most notably concerning how concepts should be defined, obtained, or learned. This has led to a rapid expansion of CBM variants and

---

*Equal contribution.
†Equal supervision.

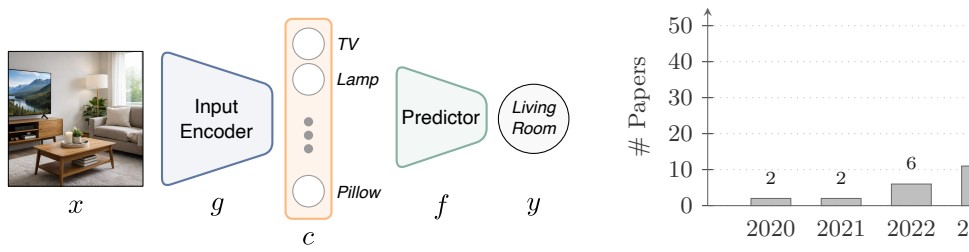

(a) Traditional schematic of CBM.

(b) Distribution of surveyed CBM papers over years.

Figure 1: **Concept Bottleneck Model (CBM) and publication trend.** (a) CBMs decompose predictions into two stages: an input encoder $g$ maps raw inputs to human-interpretable concepts $c$, and a predictor $f$ produces the final output based solely on these concepts. (b) The number of CBM papers has grown rapidly since 2020, with over 100 works published by 2025 among those surveyed in this work.

related methods in recent years (cf. Fig. 1b), further accelerated by the integration of large language and vision-language models, which have made concept extraction increasingly feasible at scale. However, this growing body of approaches lacks a unifying taxonomy: different methods tackle different subproblems, from concept discovery through intervention mechanisms to the handling of incomplete concept sets, making it difficult to understand how approaches relate to one another or to assess their respective trade-offs. Without a common framework, it remains difficult to compare methods, understand which design choices address which challenges, and identify promising directions for future research.

By asking the question *"What's in the bottle?"*, this survey aims to organize the expanding landscape of CBM research. Through a systematic and data-driven analysis of more than 100 CBM-related works, we introduce a novel taxonomy that categorizes approaches by architectural components and design choices. Our taxonomy is directly motivated by recurring challenges in the literature and reveals that the core differences and challenges in CBM approaches lie along four key architectural modules: (i) the *input module*, including input encoding and embeddings; (ii) the *concept module*, covering concept representations and the concept source (semantics and grounding); (iii) the *output module*, including the prediction model and task formulation; and (iv) the *training module*, addressing optimization strategies. Importantly, within the concept module, we identified two components that have remained largely implicit in prior work: *concept semantics* (which concepts to use and what they mean) and *concept grounding* (how concepts are detected and instantiated in data). Explicitly distinguishing these components within the concept module enables more precise comparisons across methods and clarifies design choices that were previously conflated.

Based on this taxonomy, we provide a comprehensive, structured overview of the current CBM landscape, categorizing existing methods by their architectural innovations, clarifying their assumptions, goals, and trade-offs, and highlighting structural regularities and limitations that shape the CBM design space. We systematically discuss key challenges that recent work has sought to address, from concept acquisition to intervention mechanisms, and situate CBMs within the broader context of explainable AI. Finally, we identify remaining open problems and outline promising directions for future research.

**Contributions.** This survey makes three key contributions:

- **A taxonomy grounded in architectural components.** We propose a taxonomy that systematically categorizes CBM approaches based on their design choices across four architectural modules of the CBM pipeline. Crucially, we distinguish concept semantics from concept grounding, making explicit two processes that have remained largely implicit in prior work.

- **Comprehensive categorization of existing work.** We categorize and comparatively analyze more than 100 CBM-related papers, clarifying methodological differences and situating individual contributions within a unified framework.

- **Analysis of challenges and future directions.** We identify key open problems in current approaches and outline promising research directions for improving the interpretability, reliability, and applicability of CBMs.

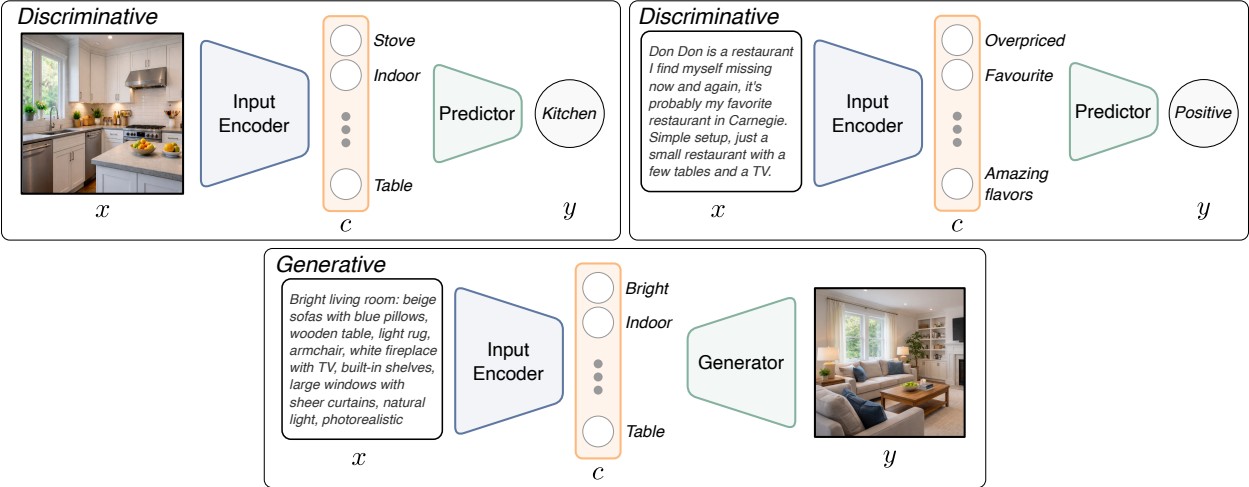

Figure 2: **Examples of CBM applications.** Top left: (i) image classification using object- or attribute-level concepts. Top right: (ii) text-based sentiment classification with semantic concepts. Bottom: (iii) a generative setup where text is mapped to concepts that condition image generation.

**Organization.** Sec. 2 provides background on CBMs and establishes the core notation. Sec. 3 reviews prior work relevant to this survey. Sec. 4 presents our taxonomy and applies it to categorize existing approaches. Sec. 5 discusses the challenges that recent works address and identifies underexplored problems. Sec. 6 outlines future research directions, connects CBMs to related areas, and Sec. 7 concludes the survey.

## 2 Concept Bottleneck Model

Concept bottleneck models address the interpretability challenge in AI by decomposing model predictions into two explicit stages: predicting human-interpretable concepts from raw inputs and then using these concepts to make the final prediction. As illustrated in Fig. 2, this architecture can be instantiated across diverse tasks, from discriminative classification to generative modeling, while maintaining the same core principle: all reasoning flows through an interpretable concept bottleneck.

A CBM is traditionally described via two modules (cf. Fig. 1a): the encoder $g$ that transforms the input into concept activations, and the predictor $f$ that predicts an output given these activations. In this work, we depart from this traditional notation to provide more detailed insights into independent design dimensions (details in Sec. 4). Formally, given an input space $\mathcal{X}$, an input embedding space $\mathcal{Z} \subseteq \mathbb{R}^D$, a semantic specification $\mathcal{S} = (s_1, ..., s_K)$ defining $K$ human-interpretable concepts with their meanings, a concept activation space $\mathcal{C} \subseteq \mathbb{R}^K$, and an output space $\mathcal{Y}$, a CBM is defined as the composition $g \circ h_{\mathcal{S}} \circ f : \mathcal{X} \to \mathcal{Y}$.

The input encoder $g : \mathcal{X} \to \mathcal{Z}$ maps the raw input $x \in \mathcal{X}$ to a $D$-dimensional input embedding $z = g(x)$. A transformation $h_{\mathcal{S}} : \mathcal{Z} \to \mathcal{C}$, conditioned on the semantic specification $\mathcal{S}$, then maps these embeddings to concept activations $c = h_{\mathcal{S}}(z)$, where the $k$-th concept activation $c_k$ corresponds to human-interpretable meaning $s_k$. Here, we explicitly distinguish between the input encoding ($g$) and the concept-discovery processes ($h_{\mathcal{S}}$). The semantic specification $\mathcal{S}$ is established through determining which concepts to use and what they mean (concept semantics), while the mapping $h_{\mathcal{S}}$ is learned through detecting these concepts in the data (concept grounding), two fundamental design choices that we elaborate on in Sec. 4. These three components, $\mathcal{S}$, $h_{\mathcal{S}}$, and $\mathcal{C}$, collectively form the *concept space*, the interpretable bottleneck through which all task-relevant information flows. Finally, the predictor $f : \mathcal{C} \to \mathcal{Y}$ produces the output $y = f(c)$ using only the concept activations. Typically, $g$ and $h_{\mathcal{S}}$ are high-capacity models (e.g., deep neural networks), while $f$ is constrained to remain interpretable, often a linear or sparse model (Koh et al., 2020). This architectural constraint enforces the bottleneck: the model cannot bypass the concept layer to encode task-relevant information directly from input to output.

This design implicitly assumes a Markovian structure where $y$ is conditionally independent of $x$ given $c$: $y \perp x \mid c$ (Havasi et al., 2022). In other words, the concepts must capture all the information about the

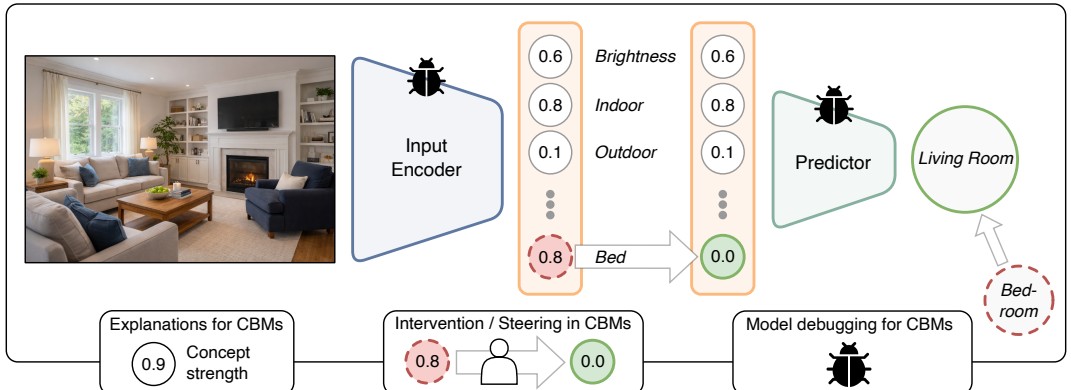

Figure 3: **Key advantages of concept bottleneck models.** CBMs provide (left) interpretable explanations via human-understandable concept strengths, (middle) interventions for correcting predictions or generating counterfactuals, and (right) structured debugging through component separation.

label present in the input. When this assumption does not hold, e.g., when the concept set is incomplete, CBMs may face limitations in predictive performance, as task-relevant information may either be lost at the bottleneck or leak through it in ways that compromise interpretability. Depending on the training strategy, $g$ and $f$ can be optimized jointly end-to-end, trained sequentially, or learned independently, each with different implications for how well the Markovian assumption is maintained in practice.

**Advantages of CBMs.** CBMs offer three key benefits that distinguish them from black-box models:

*(1) Interpretable Explanations.* Because CBMs decompose predictions through an explicit concept layer, they provide natural explanations at the level of human-understandable concepts. Thus, a user can inspect the detected concepts and understand how they contribute to the final prediction, provided that $f$ is interpretable. In Fig. 3, the model predicts the class "living room" from concepts such as high brightness (0.6) and indoor setting (0.8). Thus, the concept activations themselves serve as explanations, showing not just *what* the model predicted, but *why*.

*(2) Interventions and Steering.* A second key advantage is that the explicit concept representation provides a natural interface for human interaction. When a user detects an incorrect concept prediction, they can manually correct it and obtain an updated output prediction without retraining the model. In Fig. 3, the model incorrectly predicts the presence of a bed (0.8). A user can intervene by setting this concept to its correct value (0.0), which steers the model toward the correct prediction of "living room". This intervention capability is a central property of the CBM framework (Shin et al., 2023), enabling error correction and counterfactual reasoning ("What would the model predict if this room were less bright?"). This also relates to broader work on model steering, where users guide model behavior through high-level semantic controls.

*(3) Structured Debugging.* Finally, the structure of CBMs enables more systematic debugging than black-box models. By decomposing the full task into multiple components, practitioners can more easily isolate where failures occur. For instance, in Fig. 3, if the model mispredicts, one can determine whether the error stems from poor concept detection (e.g., incorrectly identifying the presence of a bed) or faulty reasoning over correct concepts (e.g., predicting "bedroom" despite correct concept values). This separation enables targeted improvements to specific components rather than end-to-end retraining of opaque models.

**Central Challenges in the Development of CBMs.** Although CBMs offer significant advantages and have seen considerable recent development, realizing these benefits in practice introduces specific challenges. Two central issues are: managing the interpretability-accuracy trade-off and addressing concept discovery (cf. Sec. 5.3 for a more comprehensive discussion of the challenges of CBMs).

*Interpretability-Accuracy Tradeoff.* The interpretability-accuracy trade-off describes the tension between building an expressive and performant model and maintaining the simplicity necessary for interpretability (Espinosa Zarlenga et al., 2022). At the core of this trade-off lies the Markov assumption in CBMs: the output $y$ must be conditionally independent of the input $x$ given the concept representation $c$. When

the concept set fails to encode all information required to solve the task, performance necessarily degrades compared to unconstrained black-box models. In contrast, expanding the concept space or increasing the complexity of the predictor $f$ can recover performance, but at the cost of reduced interpretability. This tension reflects the fundamental difficulty of satisfying the Markov assumption while maintaining a compact, human-interpretable concept representation.

*Concept Discovery.* The second major challenge is about the questions: where do concepts come from, and how should they be defined? Early CBM work assumed access to fully concept-annotated datasets (Koh et al., 2020; Stammer et al., 2021), an assumption that rarely holds in practice. Recent efforts have explored alternatives, such as leveraging pretrained vision-language models to obtain concept annotations (Oikarinen et al., 2023; Yang et al., 2023) or discovering concepts directly from data in an unsupervised manner (Rao et al., 2024). Regardless of the approach, developing CBMs requires a concept set that is both sufficient to solve the task and small enough for users to meaningfully inspect and understand. Balancing these requirements through supervised or unsupervised methods remains an active and challenging area of research.

**Relations to other Concept-based Approaches.** To clarify what distinguishes CBMs, it is helpful to contrast them with related concept-based methods that differ in when and how concepts are used. Several approaches use concepts for *post-hoc explanation* of trained models. Concept Activation Vectors (CAVs) (Kim et al., 2018; Ghorbani et al., 2019) identify directions in a model's latent space corresponding to human-interpretable concepts, enabling analysis of how sensitive predictions are to those concepts. Unlike CBMs, CAVs do not constrain the model architecture or training. They are used as analytical tools applied after the training to test for learned concepts. Other approaches integrate concepts *by design* into the model architecture, though with mechanisms different from those of CBMs. Before the deep learning era, attribute-based methods (Lampert et al., 2009; Kumar et al., 2011) used human-defined attributes as intermediate representations for zero-shot classification. More recently, prototype networks (ProtoPNets) (Chen et al., 2019; Bontempelli et al., 2022) base predictions on learned prototypical examples rather than explicit concept predictions, Self-Explaining Neural Networks (SENNs) (Alvarez-Melis & Jaakkola, 2018) learn basis concepts without requiring human-semantic alignment, and sparse autoencoders (SAEs) (Templeton et al., 2024) discover high-dimensional sparse features post-hoc rather than enforcing a low-dimensional concept bottleneck during training. All of these approaches differ from CBMs in either the nature of their intermediate representations or how concepts are integrated into the architecture. CBMs also connect to broader research areas, including neuro-symbolic AI, compositional scene understanding, and structured prediction. We discuss these relationships in detail in Sec. 6.2. Overall, the idea of human-understandable concepts as a building block of ML models has been and continues to be an important direction in the context of explainable and interpretable machine learning. We organize these extensions by increasing departure from the original formulation, noting that each modifies one or more of the four modules defined in our taxonomy.

# 3    Related Work

The CBM literature has expanded rapidly and diversified in many directions, yet systematic efforts to organize and categorize recent research remain limited.

*CBM-focused Surveys.* Mpinda et al. (2026) survey 17 CBM with a focus on applicability to medical data, distinguishing them by task type (single- vs. multi-label) and datasets employed, but do not provide a detailed architectural taxonomy. Wang et al. (2026) present a survey of 36 CBM works organized in four high-level dimensions: concept acquisition, concept-based decision making, interventions, and evaluation. Although this categorization captures some aspects of CBM design, it mixes architectural components with challenges and evaluation criteria, making it difficult to independently analyze choices across these dimensions.

In contrast, our work provides a fine-grained architectural taxonomy grounded in a systematic review of over 100 CBM works. The taxonomy distinguishes CBMs based on their three main architectural elements, namely the input, concept, and output modules, as well as their training procedure. This study is the first to include all architectural aspects of a CBM in a taxonomy, rather than focusing exclusively on the concept module. Furthermore, the proposed taxonomy separates previously entangled elements, such as concept acquisition, into distinct categories, thereby enabling a structured and meaningful distinction among prior CBM works.

We further separate architectural design choices from the challenges they aim to address, enabling a clear comparison of existing methods and providing a structured basis for identifying open problems.

*Concept-based XAI Surveys.* Several surveys on the broader field of concept-based explainability touch upon CBMs as one approach among many. Poeta et al. (2023) provide a comprehensive overview of concept-based XAI, covering both post-hoc methods (e.g., CAVs (Kim et al., 2018)) and interpretable-by-design models, including CBMs, but do not categorize CBM architectures in detail. Similarly, Alharith et al. (2026) survey concept-based XAI with only brief mentions of specific CBM works. Lee et al. (2025a) focus specifically on extracting concepts from pretrained neural networks through post-hoc analysis, but do not cover interpretable-by-design architectures like CBMs.

*Related Areas.* Surveys in adjacent fields also intersect with CBMs. Bhuyan et al. (2024) survey neuro-symbolic AI and identify challenges such as concept-symbol alignment, but cover concept-based models only briefly. Gupta & Narayanan (2024) and Teso et al. (2023) partially survey concept-based model improvement and interactive learning; while many covered methods apply to CBMs, neither provides a detailed CBM-specific analysis.

*Position Papers and Challenges.* Several works identify open problems in concept-based approaches. Barbiero et al. (2025) formalize interpretability as inference equivariance and derive foundational principles that also apply to CBMs. Lee et al. (2024) give an overview of concept-based XAI and possible avenues for future advancements. Some challenges also apply to CBMs, namely, concepts for temporal and multimodal settings and probabilistic representation of concepts. Sinha & Zhang (2025) collect risks and limitations of concept-based models. They mention concept leakage, spatial grounding of concepts, and (lack of) robustness to input perturbations as key challenges. While these papers mention some current limitations of CBMs, our survey of the field's current state revealed additional challenges and directions that have not been uncovered.

In contrast to these related surveys, this work provides a unified taxonomy of CBMs that is more detailed and comprehensive than previous attempts. In addition, we provide an extensive survey of the current state of the field, leading to a data-driven analysis and collection of open issues and future challenges to tackle.

# 4 Taxonomy

In this section, we introduce a taxonomy of concept bottleneck models (CBMs) that follows the modular structure of CBM architectures. The taxonomy (cf. Fig. 4 (top)) is grouped around the core modules of a CBM (formally introduced in Sec. 2). The **input module** comprises the *input modality*, the *input encoder* $g(\cdot)$, and the resulting *input embeddings z*. The **concept module** $h(\cdot)$ is divided into the *concept source*, which specifies what and how concept information is obtained, and the *concept activations c*, which represent concepts at the sample level. Importantly, within the concept source, we distinguish two related processes: *concept semantics*, which defines the set of task-relevant concepts and the corresponding human-interpretable semantics, and *concept grounding*, which links these abstract concepts to patterns in the input data. Next, the **output module** consists of the *predictor* $f(\cdot)$ and the type of *task* addressed by the CBM. Finally, **training** covers the general training strategy used. A detailed breakdown of common component types is provided in Fig. 4 (bottom).

The proposed taxonomy is derived from a data-driven analysis of the surveyed literature and is designed to be general and extensible, allowing it to accommodate future developments in the field. In the following, we discuss each component individually, organized by its corresponding module. Throughout, we provide representative references from the complete list of reviewed works in Tab. 1 to illustrate different instantiations. The section concludes with a brief summary of observed trends.

## 4.1 Input Module

The first of the four concept bottleneck modules focuses on how input data is processed before entering the concept space. It encompasses the *input modality*, the choice of *input encoder*, and the structure of the resulting *embedding* representation, defining the foundation upon which the concept bottleneck is constructed.

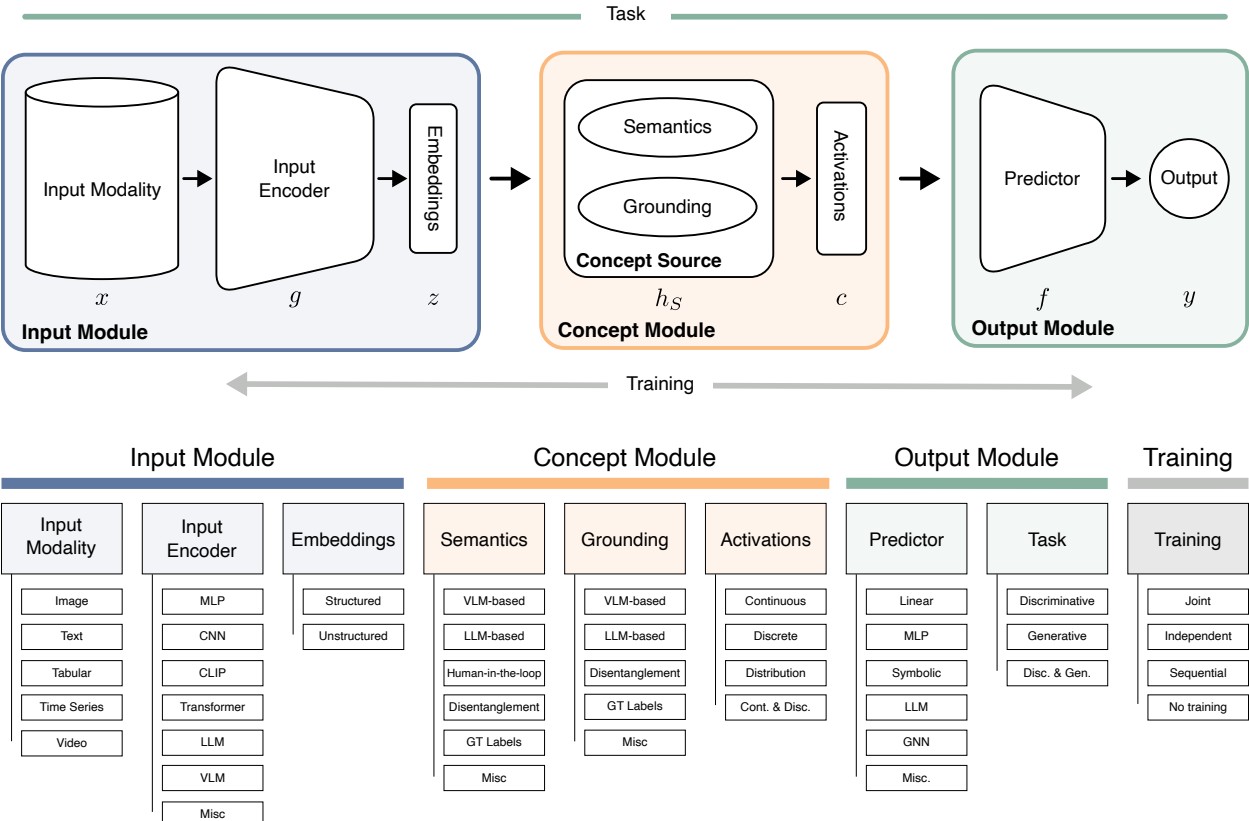

Figure 4: **Overview of our proposed taxonomy of CBMs. (Top)** Next to more general components like the *input modality*, the *training* and the *task*, the remaining components cover the typical CBM architecture: The *input encoder g*, its *embeddings z*, the *predictor f* and the concept module, split into *concept semantics* (which concepts and what do they mean?) and *concept grounding* (how do we connect these to input patterns?), as well as *concept activations* (final concept representation). **(Bottom)** Details on the taxonomy. This schema illustrates the common types each component can take, as found by our literature review.

### 4.1.1 Input Modality

The first component of our taxonomy concerns the modality of the data (input space $\mathcal{X}$) on which the CBM operates. Although this aspect is not an architectural element of the model itself, it fundamentally shapes all subsequent design choices. The data modality determines how inputs must be encoded and constrains what can meaningfully serve as a concept. What qualifies as a concept varies substantially across domains; for instance, attributes such as color are meaningful in image data but have no direct counterpart in time-series settings. The recent research on CBMs has largely been focused on *images* as input data, with more than 80% of the papers exclusively focusing on this modality. However, some works have branched out towards *text* (Sun et al., 2025; Bhan et al., 2025), *tabular data* (Espinosa Zarlenga et al., 2023c; Vandenhirtz et al., 2024), *video* (Knab et al., 2025b; Lee et al., 2025b), and *time series* data (Wu et al., 2022; van Sprang et al., 2024). Notably, our categorization reflects the input modalities used in each work's original experiments, though some approaches may be more broadly applicable across modalities than their presentation suggests. In summary, the types of input modalities identified over all surveyed CBM approaches of Tab. 1 are: *image*, *text*, *tabular*, *time series*, or *video*.

### 4.1.2 Input Encoder

The input encoder $g(\cdot)$ transforms the input data into an embedding space suitable for extracting or identifying relevant concepts. We hereby distinguish between several different encoder types that are frequently used in CBMs. Some CBMs use a fully trained *MLP* as input encoder, in particular for more structured data

(Debot et al., 2024; Espinosa Zarlenga et al., 2023c; 2022). For vision data, *CNNs* like ResNet (He et al., 2016) or *transformers* like ViT (Dosovitskiy, 2020) are often employed as backbone architectures. Pretraining these models on popular image datasets allows to build on a base feature structure and rich embeddings when learning to represent concepts (Koh et al., 2020; Sawada & Nakamura, 2022; Hu et al., 2025). To profit even more from the power of pretrained models, many recent CBMs build upon *CLIP* (Radford et al., 2021), or similar powerful language and text encoders (Oikarinen et al., 2023; Yang et al., 2023; Rao et al., 2024; Bhalla et al., 2024), in particular when limited concept supervision is available. When the input data comes from a specific domain, such as medical images or videos, specialized pretrained models are often utilized that suit these modalities (Knab et al., 2025b; Mpinda et al., 2026; Gao et al., 2024b). In summary, the input encoders identified in the approaches include *MLP*, *CNN*, *CLIP*-like models, *Transformers*, *LLMs*, *VLMs* and a *miscellaneous* category.

### 4.1.3 Input Embeddings

The input embedding $z$ generated by the input encoder serves as the first information bottleneck in a CBM, determining which information from the raw input is preserved for concept extraction. Information lost at this stage cannot be recovered by subsequent components. While embeddings depend on the encoder architecture, their *structure* determines the type and granularity of extractable concepts. We therefore ask: *Does the embedding contain inherent structure?* In most existing work, input embeddings are flat, continuous vectors (Koh et al., 2020; Yan et al., 2023). Within this unstructured vector, all information is entangled, and concepts must be extracted globally from the entire representation, typically supporting only image-level or global concepts. In contrast, alternative approaches utilize *structured* embeddings which enable localized concept grounding (e.g., tying concepts to specific input regions or objects), support compositional reasoning (concepts can be composed across structured elements), and allow for more targeted interventions (modifying concepts for specific components rather than globally). Existing approaches in the visual domain split the input image into slots or objects (Stammer et al., 2021; Hong et al., 2024; Steinmann et al., 2025), patches (Panousis et al., 2024), or segmentations (Eisenberg et al., 2025). Other works utilize graph-structured embeddings to represent causal (De Felice et al., 2025) or relational (Barbiero et al., 2024; Xu et al., 2025) dependencies, enabling more explicit relational reasoning. Further approaches impose structure by decomposing embeddings into interpretable and residual (opaque) factors (Marconato et al., 2022; Martínez-García et al., 2026), enabling explicit control over information that is not captured by predefined concepts. Consequently, the input embedding category is divided into *structured* and *unstructured* embeddings.

Table 1: **Categorization of CBM approaches in our taxonomy.** Some components are abbreviated for readability, while others are spelled out. Abbreviations used: Modality: Image, Tabular, Time Series; Encoder: Transformer-based, Miscellaneous; Embedding (Emb.): Structured / Unstructured; Semantics & Grounding: Dictionary learning, Ground Truth, Disentanglement, Human-in-the-loop; Concept Activation (C-Act.): Continuous, Discrete, Distributional; Task: Discriminative / Generative; Training: Joint / Sequential / Independent / – (No training required).

| Publication | Modality | Encoder | Emb. | Semantics | Ground. | C-Act. | Pred. | Task | Training |
|---|---|---|---|---|---|---|---|---|---|
| Koh et al. (2020) | Img | CNN | Unstr | GT | GT | Cont | Linear | D | J, S, I |
| Lage & Doshi-Velez (2020) | Tab | MLP | Unstr | Human | Misc | Disc | Linear | D | J |
| Losch et al. (2021) | Img | CNN | Unstr | GT, Dis | GT, Dis | Cont, Disc | MLP | D | S |
| Stammer et al. (2021) | Img | CNN | Str | GT | GT, Dis | Disc | MLP | D | I |
| Wu et al. (2022) | TS | Misc | Str | GT | GT | Cont | Linear | D | I |
| Sawada & Nakamura (2022) | Img | CNN | Unstr | GT | GT | Cont | Linear | D | J |
| Espinosa Zarlenga et al. (2022) | Img | CNN | Unstr | GT | GT | Cont | Linear | D | J |
| Havasi et al. (2022) | Tab, Img | CNN, MLP | Unstr | GT | GT | Cont | MLP | D | I |
| Stammer et al. (2022) | Img | CNN | Unstr | Human | Dis | Disc | MLP | G | J |
| Marconato et al. (2022) | Img | CNN | Str | GT | GT, Dis | Distr | Linear | D | J |
| Wang et al. (2023) | Img | CNN | Unstr | Human | Dis | Cont | Linear | D | J |
| Yuksekgonul et al. (2023) | Img | Misc | Unstr | GT, VLM | Dis | Cont | Linear | D | S |
| Espinosa Zarlenga et al. (2023c) | Tab | MLP | Unstr | GT | Dis | Cont | Linear | D | J |
| Espinosa Zarlenga et al. (2023b) | Img | CNN | Unstr | GT | GT | Cont | MLP | D | S |
| Shin et al. (2023) | Img | CNN | Unstr | GT | GT | Cont | Linear | D | J, S, I |
| Kim et al. (2023b) | Img | CLIP | Unstr | LLM | LLM | Cont | Linear | D | S |
| Yang et al. (2023) | Img | CLIP | Unstr | LLM | VLM | Cont | Linear | D | S |
| Oikarinen et al. (2023) | Img | Misc | Unstr | LLM | VLM | Cont | Linear | D | J |
| Yan et al. (2023) | Img | CLIP | Unstr | LLM | VLM | Cont | Linear | D | J, I |

| Publication | Domain | Encoder | Emb. | Semantics | Ground. | C-Act. | Pred. | Task | Training |
|---|---|---|---|---|---|---|---|---|---|
| Chauhan et al. (2023) | Img | CNN | Unstr | GT | GT | Cont | MLP | D | J, I |
| Kim et al. (2023a) | Img | CNN | Unstr | GT | GT | Distr | Linear | D | S |
| Barbiero et al. (2024) | Tab, Img | Misc | Str | GT | GT | Cont | GNN | D | J |
| Hu et al. (2024a) | Img | CNN | Unstr | VLM | VLM | Cont | Symbolic | D | J |
| Huang et al. (2024) | Img | CNN, Trans. | Unstr | GT | GT | Cont | Linear | D | J |
| Lai et al. (2024) | Img | CLIP | Unstr | LLM | VLM | Cont | Linear | D | S |
| Kim & Ko (2024) | Img | CNN | Str | GT | GT | Cont | GNN | D | J, I |
| Rao et al. (2024) | Img | CLIP | Unstr | VLM | Dis | Cont | Linear | D | S |
| Steinmann et al. (2024b) | Img | CNN | Unstr | GT | GT | Cont | Linear | D | I |
| Hu et al. (2024b) | Img | CNN | Unstr | GT | GT | Cont | Linear | D | S |
| Hong et al. (2024) | Img | CNN, Trans. | Str | GT, Dis | GT, Dis | Cont | Linear | D | J |
| Singhi et al. (2024) | Img | CNN | Unstr | Human | Misc | Cont | Linear | D | J, S, I |
| Chowdhury et al. (2024) | Img | CLIP | Unstr | VLM | VLM | Cont | Linear | D | S |
| Srivastava et al. (2024) | Img | CLIP | Unstr | VLM | VLM | Cont | Linear | D | S |
| Tan et al. (2024a) | Img | CLIP, CNN | Unstr | VLM | VLM | Cont | Linear | D | J |
| Tan et al. (2024b) | Text | LLM | Unstr | GT, VLM | GT, VLM | Distr | Linear | D | J |
| Shang et al. (2024) | Img | CLIP | Unstr | LLM | VLM | Cont | Linear | D | S |
| Wang et al. (2024) | Img | CNN | Unstr | LLM | VLM | Cont | Linear | D | J |
| Jeon et al. (2024) | Img | CLIP | Str | LLM | VLM | Cont | Linear | D | J |
| Bhalla et al. (2024) | Img | CLIP | Unstr | GT | VLM | Cont | Linear | D | S |
| Panousis et al. (2024) | Img | CLIP | Str | GT | VLM | Cont | Linear | D | S |
| Laguna et al. (2024) | Img | CNN | Unstr | GT | GT | Cont | Misc | D | J |
| Ismail et al. (2024) | Img | Misc | Unstr | GT | GT | Cont | Misc | G | J |
| Xu et al. (2024) | Img | CNN | Unstr | GT | GT | Cont | Misc | D | J |
| Debot et al. (2024) | Img | MLP | Unstr | GT | GT | Cont | Symbolic | D | J |
| Lorello et al. (2024) | Img | CNN, Trans. | Unstr | Misc | Misc | Cont, Disc | Linear | D | S |
| Stammer et al. (2024) | Img | CNN, MLP | Str | Human, Dis | Misc | Cont, Disc | Misc | D | S |
| Delfosse et al. (2024) | Img | CNN, MLP | Str | GT, VLM | GT, VLM | Disc | Symbolic | D | S |
| Parekh et al. (2024) | Img | CNN | Unstr | Misc | Misc | Cont | Linear | D | J |
| Vandenhirtz et al. (2024) | Tab, Img | CNN | Unstr | GT | GT | Distr | Linear | D | J |
| Kim et al. (2024) | Img | CNN | Unstr | GT | GT, Dis | Distr | Linear | D | J |
| van Sprang et al. (2024) | TS | Trans. | Unstr | GT | GT | Cont | Misc | D | J |
| Gao et al. (2024a) | Img | CNN, Trans. | Unstr | GT | VLM | Cont | Linear, MLP | D | J |
| Gao et al. (2024b) | Img | CLIP | Unstr | GT, LLM, Dis | VLM | Cont | Linear | D | J |
| Kazmierczak et al. (2024) | Img | CLIP | Unstr | LLM | VLM | Cont | Misc | D | – |
| Xu et al. (2025) | Img | CLIP | Str | LLM | VLM | Cont | GNN | D, G | S |
| Lee et al. (2025c) | Video | Trans. | Unstr | Misc | Misc | Cont | Linear | D | S |
| Schrodi et al. (2025) | Img | CLIP | Unstr | LLM | Dis | Cont | Linear | D | S |
| Kulkarni et al. (2025a) | Img | CLIP | Unstr | GT | Dis | Cont | Linear | D | J |
| Kulkarni et al. (2025b) | Img | DNN | Unstr | GT | GT, VLM | Cont, Disc | MLP | G | S |
| He et al. (2025a) | Img | CNN | Unstr | Human | GT | Cont | Linear | D | J |
| Huy et al. (2025) | Img | CNN | Unstr | GT | GT | Cont | Linear | D | S |
| Hu et al. (2025) | Img | CNN | Unstr | GT | GT | Cont | Linear | D | J |
| Park et al. (2025) | Img | CNN, Trans. | Unstr | GT | GT | Cont | Linear | D | S |
| Zhang et al. (2025) | Img | CNN | Unstr | GT | GT | Cont | Linear | D | S |
| Ismail et al. (2025b) | Text | LLM | Unstr | GT | GT | Cont | Linear | G | I |
| Bader et al. (2025) | Img | CNN | Unstr | VLM | Misc | Cont | Linear | D, G | J, I |
| Penaloza et al. (2025) | Img | CNN | Unstr | GT | GT | Cont | Linear | D | J |
| Sun et al. (2025) | Text | LLM | Unstr | LLM | LLM | Cont | Linear | D, G | S |
| Eisenberg et al. (2025) | Img | CLIP | Str | VLM | VLM | Cont | Linear | D | S |
| Choi et al. (2025) | Img | CLIP | Unstr | VLM | VLM | Cont | Linear | D | S |
| Prasse et al. (2025) | Img | CLIP | Unstr | VLM | VLM | Cont | Linear | D | S |
| Lee et al. (2025b) | Video | Trans. | Unstr | VLM | VLM | Cont | Linear | D | S |
| Zhao et al. (2025) | Img | CLIP | Unstr | VLM | VLM | Cont | Linear | D | S |
| Yamaguchi et al. (2025) | Img | CNN | Unstr | VLM | VLM | Cont | Linear | D | – |
| Steinmann et al. (2025) | Img | CLIP | Str | LLM | VLM | Cont | Linear | D | S |
| Alam et al. (2025) | Img | VLM | Unstr | LLM | VLM | Cont | Linear | D, G | S |
| Lin et al. (2025) | Img | CLIP | Unstr | GT, LLM | GT, VLM | Cont | Misc | D | J |
| Liu et al. (2025) | Img | CLIP | Unstr | LLM | VLM | Cont | Linear | D | S |
| Benou & Raviv (2025) | Img | CLIP | Unstr | LLM | VLM | Cont | Linear | D | S |
| Mehra et al. (2025) | Img | CLIP | Unstr | LLM | VLM | Cont | Linear | D | S |
| He et al. (2025b) | Img | CNN, CLIP | Unstr | GT, VLM | GT, VLM | Cont | LLM | D | I |
| He et al. (2025c) | Img | CLIP | Unstr | Misc | VLM | Cont | Linear | D | S |
| Puri et al. (2025) | Text | LLM | Unstr | GT | GT | Cont | MLP | D | I |
| Kalampalikis et al. (2025) | Img | CNN, Trans. | Unstr | GT | GT | Cont | MLP | D | J |
| Stropeni et al. (2025) | Img | CNN | Unstr | VLM | VLM | Cont | MLP | D | J, S, I |
| Yamaguchi & Nishida (2025) | Img | LLM, Trans. | Unstr | VLM | VLM | Cont | MLP | D | J |
| Pugnana et al. (2025) | Img | Misc | Unstr | GT | GT | Cont | MLP | D | I |
| Vemuri et al. (2025) | Img | CNN | Unstr | GT, LLM | GT | Cont | Symbolic | D | J |
| De Felice et al. (2025) | Img | CNN, MLP | Str | LLM | VLM | Cont | Symbolic | D | J |
| Almudévar et al. (2025) | Img | CNN | Unstr | GT | GT | Cont, Disc | Linear | D | J |
| Feng et al. (2025) | Tab, Img, Text | Misc | Unstr | VLM | VLM | Cont, Disc | Linear | D | S |
| Jiang et al. (2025) | Img | CLIP | Unstr | LLM | VLM | Cont, Disc | Linear | D | J |
| Wang et al. (2025) | Img | CLIP | Unstr | GT | VLM | Cont | Linear | D | J |
| De Santis et al. (2025) | Img | CNN, Trans. | Unstr | Dis | Dis | Disc | Linear | D | J |
| Dominici et al. (2025) | Img | CLIP | Unstr | GT | GT, VLM | Disc | Linear | D | S |
| Bhan et al. (2025) | Text | LLM | Unstr | LLM | LLM | Disc | Linear | D | J, S, I |
| Patrício et al. (2025a) | Img | VLM | Unstr | VLM | VLM | Disc | LLM | D | – |
| Patrício et al. (2025b) | Img | VLM | Unstr | GT, VLM | VLM | Disc | LLM | D | – |
| Parchami-Araghi et al. (2025) | Img | CNN, Trans. | Str | Dis | Dis | Cont | Linear | D | S |

| Publication | Domain | Encoder | Emb. | Semantics | Ground. | C-Act. | Pred. | Task | Training |
|---|---|---|---|---|---|---|---|---|---|
| van Krieken et al. (2025) | Img | DNN | Unstr | GT | Dis | Disc | Symbolic | D | J |
| Knab et al. (2025b) | Video | CLIP | Unstr | VLM | VLM | Cont | Misc | D | S |
| Du et al. (2025) | Img | CNN, CLIP | Unstr | LLM | VLM | Cont | MLP | D | S |
| Turan et al. (2025) | Img | CNN | Unstr | LLM, Human | VLM, Dis | Disc | MLP | D | S |
| Asiyabi et al. (2026) | Tab, Img | Misc | Str | GT | GT | Cont | MLP | D | I |
| Mpinda et al. (2026) | Img | CNN, VLM, CLIP | Unstr | LLM | VLM | Cont | MLP | D | S |
| Wittenmayer et al. (2026) | Img | CLIP, Trans. | Str | VLM | Dis | Cont | Linear | D, G | S |
| Zhang et al. (2026) | Img | Misc | Str | VLM | VLM | Cont | Linear | D | J |
| Martínez-García et al. (2026) | Img | Misc | Str | GT | GT, VLM | Cont, Disc | Misc | G | S |

## 4.2 Concept Module

The concept module serves as the framework's bottleneck and is its central element. Its role is to map the input embeddings $z$ into human-interpretable *concept activations c* via the transformation $h_\mathcal{S}$. It comprises three interrelated components: the *concept source*, which is further subdivided into *semantics* and *grounding*, and the resulting *concept activations* themselves, together constituting the concept space.

### 4.2.1 Concept Source

A central design choice of CBMs is clarifying the *source* of concepts, i.e., where concept information comes from and how it is obtained. Existing approaches span a wide spectrum, ranging from fully supervised formulations to weakly supervised and fully unsupervised methods.

Importantly, obtaining concepts actually involves **two distinct but interrelated processes**. **Concept semantics** (*Which concepts should the model use, and what do they mean?*) determines which concepts are relevant for the task and what they mean semantically, establishing the concept vocabulary $\mathcal{S}$ and its human-interpretable semantics. **Concept grounding** (*How do we connect abstract concepts to perceptual patterns?*) addresses how abstract concepts acquire operational meaning by connecting to patterns in the input data $h_\mathcal{S}$. This includes recognizing concept *presence* in inputs and, *localizing* these to specific input regions (relating to the fundamental symbol grounding problem (Harnad, 1990); cf. Sec. 6.1). The output of concept semantics is a concept vocabulary with semantic descriptions that humans can understand and reason about (obtaining $\mathcal{S}$). The output of concept grounding is a learned mapping from inputs to concept activations (i.e., obtaining $h_\mathcal{S}$), enabling the model to operationally detect concepts in new data. These processes are essential, but address fundamentally different questions and can proceed in different orders, leading to distinct methodological differences, as described below. Although analytically separable, some implementations realize both components jointly or implicitly.

**Top-Down: Semantics Precedes Grounding.** *Start with semantic concepts, then learn to detect them.* In this workflow, concepts are first defined, their semantic meaning and relevance established, before being grounded in perceptual data (i.e., starting with $\mathcal{S}$ and then learning the mapping $h_\mathcal{S}$ afterward). The concept semantics phase determines the vocabulary through various means: many early works assume access to complete, expert-annotated concept sets (Koh et al., 2020; Sawada & Nakamura, 2022), where domain experts specify which concepts matter and what they mean. Following the emergence of capable LLMs and VLMs, several works leverage the knowledge embedded in these pretrained models to generate concept vocabularies automatically (Yang et al., 2023; Hu et al., 2024a), utilizing their language understanding to propose task-relevant concepts without additional human annotations. Once concepts are defined in this top-down manner, the concept grounding phase learns to detect them in the data. Many supervised CBMs hereby rely on concept annotations and train their input encoders to directly predict concept presence through standard supervised learning (Stammer et al., 2021; Barbiero et al., 2024), where grounding emerges as the encoder learns to associate input features with concept labels. A different approach leverages pretrained vision-language models such as CLIP (Radford et al., 2021), e.g., using cosine similarity between image embeddings and textual concept embeddings (Oikarinen et al., 2023; Bhalla et al., 2024; Chowdhury et al., 2024), or between image embeddings and prototypical segmented image embeddings (Prasse et al., 2025) as grounding signals. Such forms of grounding exploit the semantic alignment from pretraining, e.g., by matching images with textual descriptions. Overall, this top-down workflow is predominant when substantial prior knowledge exists about task-relevant concepts, either from domain experts or foundation models.

**Bottom-Up: Grounding Precedes Semantics.** *Discover patterns in data, then assign semantic meaning.* In this workflow, the order is reversed: first, patterns are found in perceptual data through unsupervised discovery, then subsequently given human-understandable semantics. The concept grounding phase employs unsupervised methods to discover recurring patterns or latent factors in the input space, grounding them to abstract "raw" concepts (creating an initial mapping $h$) before they receive human semantic interpretations (finding $\mathcal{S}$ and effectively transforming the mapping to $h_{\mathcal{S}}$). Disentanglement methods (Rao et al., 2024; Yuksekgonul et al., 2023; Espinosa Zarlenga et al., 2023c; De Santis et al., 2025) learn to separate independent factors of variation, grounding each factor to distinct data characteristics such as color, shape, or texture. Once the patterns are grounded, the concept semantics phase aligns these discovered patterns with human understanding. This involves assigning interpretable names to grounded factors, merging redundant patterns, or refining concept granularity to match human concepts. Post-hoc semantics is often performed interactively with users who examine the grounded patterns and provide semantic labels (Wang et al., 2023; Stammer et al., 2024) (human-in-the-loop). Alternatively, LLMs and VLMs can generate natural language descriptions for discovered patterns (Rao et al., 2024). This bottom-up workflow is particularly valuable when limited prior knowledge exists about which concepts are relevant, allowing data-driven discovery to reveal task-relevant structure that might not be anticipated by domain experts.

To summarize, the semantics category comprises *VLM-based*, *LLM-based*, *human-in-the-loop* methods, *disentanglement* techniques, and *ground-truth labels*. The grounding category similarly includes *VLM-based*, *LLM-based*, *disentanglement*, and *ground-truth labels*. Both components contain a *miscellaneous* group.

### 4.2.2 Concept Activations

Through the concept grounding process, the model produces concept activations $c$, i.e., numerical representations that encode the presence, absence, or degree thereof for each concept in a given input. These activations form the explicit bottleneck through which all task-relevant information must flow, making this representation a critical design choice that affects several properties of a model, such as interpretability, intervention effectiveness, and susceptibility to concept leakage (cf. Sec. 5.3). The majority of existing works use *continuous* activations, where each concept is represented by a real-valued *score* (Koh et al., 2020; Yang et al., 2023). Continuous activations typically lie in $[0, 1]$, where values near 0 indicate absence, values near 1 indicate presence, and intermediate values capture uncertainty or partial presence of a concept. In principle, however, continuous activations may take any real value depending on a concept's semantic range. To address issues of continuous values (cf. Sec. 5.3), some works specifically employ *discrete* activations (Feng et al., 2025; De Santis et al., 2025; Patrício et al., 2025b), where concepts take on categorical values. A third option observed across the field is *distributional* activations (Kim et al., 2023a; Vandenhirtz et al., 2024), where each concept is represented as a probability distribution over its possible values rather than a single point estimate. As the choice of activation representation involves trade-offs in different areas, some recent works started to combine multiple representation forms, e.g., maintaining both continuous and discrete representations (Stammer et al., 2024; Patrício et al., 2025b). In summary, the concept activation component, encompasses *continuous*, *discrete*, *distributional*, and hybrid (*continuous & discrete*) representations.

## 4.3 Output Module

Once the concept activations are obtained, they serve as the explicit interface for human inspection and interaction, and are processed by the output module through a *predictor* to produce *task*-specific predictions.

### 4.3.1 Predictor

The predictor is the last architectural component of a CBM in the taxonomy and is typically denoted by $f$. It generates a task prediction based on the concept activations $c$. To preserve the overall interpretability of the CBM, the predictor should ideally be an interpretable model, ensuring users can understand not only which concepts were detected but also how they informed the final decision. Various predictor models have been proposed in existing literature, which we categorize into the following types. *Linear layers* are by far the most popular predictor type (Koh et al., 2020; Espinosa Zarlenga et al., 2022; Sun et al., 2025). To be able to model more complex dependencies between concepts and outputs, several CBMs have also

referred to forms of *symbolic reasoning* (Vemuri et al., 2025; Debot et al., 2024) or *LLM-based reasoning* (He et al., 2025b; Patrício et al., 2025a). If the concepts have a graph structure, a *GNN* is commonly used to make predictions based on the corresponding "concept graph" (Barbiero et al., 2024; Xu et al., 2025). Lastly, several papers also use an *MLP* or other shallow, but less interpretable models (Chauhan et al., 2023; Pugnana et al., 2025), which might increase performance but come at the cost of reduced interpretability. Accordingly, the predictor category includes *linear* models, *MLPs*, *symbolic* models, *LLM*-based predictors, *GNNs*, as well as a *miscellaneous* group.

### 4.3.2 Task

As the *task* a model should solve can affect all of its elements, it is considered a component of this taxonomy, even if it is not directly part of the architecture. While there are many different tasks for which CBMs have been used, here we distinguish mainly between two high-level categories: *discriminative* and *generative* tasks. A discriminative task aims to model the conditional relationship between concept activations and the output, i.e., $P(y \mid c)$, or more generally, a direct mapping $f(c) \to y$, such as predicting the probability of a living room given that it is indoors, has a table, and a TV. On the other hand, a generative task aims to model the joint distribution of concepts and outputs $P(C, Y)$ and to generate new samples from it, such as a new image of a living room with cozy chairs and a sofa. Nearly all current works on CBMs focus on discriminative tasks, primarily single-label classification. However, some notable works have begun to explore generative tasks (Ismail et al., 2024; Kulkarni et al., 2025b). The task component, therefore, comprises *discriminative* approaches, *generative* approaches, and hybrid settings in which *discriminative & generative* objectives are jointly addressed within the same model.

### 4.4 Training Module

Finally, while the exact training details of a CBM depend on the choices for its individual components, we consider it to be a component not specifically tied to any of the previous three modules (input, concept, output) in particular. The general training strategy can be split into three categories: *Joint* training refers to models that jointly train the input encoder and the predictor, and is among the most commonly adopted training strategies. This allows all components to be tuned to work well together, but can also lead to problems such as concept leakage (Havasi et al., 2022). *Sequential* training refers to first training the input encoder and then training the predictor on the predicted concepts while keeping the encoder frozen, and is likewise widely used in the literature. This can reduce concept leakage, but can still result in unintuitive intervention behavior, as a user does not necessarily know what concept activations the encoder typically produces (Shin et al., 2023). During *independent* training, the input encoder and predictor are independently trained. This approach is usually desirable to reduce leakage and for effective intervention mechanisms, but potentially requires access to concept labels when training both parts of the model, which is often a limitation in practice (Margeloiu et al., 2021). In summary, the training dimension of the taxonomy comprises *joint*, *independent*, and *sequential* training strategies, as well as settings in which *no training* is required.

### 4.5 Summary of Observed Trends

Building on Tab. 1, we observe several consistent patterns across the CBM literature. There is a *strong overrepresentation of image-based approaches*, while other modalities such as text, tabular data, or video remain comparatively underexplored. Recent work increasingly relies on foundation models, in particular *CLIP as an encoder* and *LLMs/VLMs for concept semantics and grounding*, reflecting a shift toward scalable, weakly supervised pipelines, often combined with *unstructured embeddings*. Furthermore, *continuous concept activations clearly dominate*, whereas probabilistic or distributional representations remain underrepresented despite their advantages for uncertainty modeling. The output module is largely characterized by *linear predictors*, with MLPs as a common fallback when higher expressiveness is required, while more advanced reasoning modules remain rare. In addition, most CBMs focus on *discriminative tasks*, with only a small but growing number of works addressing generative settings. Finally, *training strategies remain heterogeneous*, with no dominant paradigm emerging.

Overall, these observed trends highlight structural biases in current CBM design and directly motivate the challenges and future directions discussed in the following sections.

## 5 Challenges and Open Problems in CBMs

Building on the taxonomy introduced in Sec. 4, we now move from describing design dimensions to analyzing the practical and theoretical challenges that arise when constructing and deploying CBMs. We organize the discussion into three categories: challenges related to (i) the design of the concept module, (ii) the model reliability, and (iii) validation and scope. For each category, we summarize the current progress and highlight representative approaches that address these issues. This perspective complements the taxonomy by explaining the motivations behind many of the architectural variations identified earlier.

### 5.1 Concept Module Design and Representational Challenges

The concept module lies at the heart of any CBM, and consequently, the main challenge is to obtain and represent concept information effectively. Several further challenges arise around this main question, for example, which concepts should be included in $\mathcal{C}$, and how to ensure they are properly grounded. We therefore outline the specific facets of this underlying challenge in the following paragraphs.

**Representing Concepts.** Finding an appropriate concept representation is challenging: it must be sufficiently expressive to capture task-relevant structure while remaining human-understandable, and stable under distribution shifts. Within the concept module, several forms of concept activations have been proposed to address these competing desiderata. Continuous-valued vectors enable smooth optimization and can capture graded or compositional semantics (Yuksekgonul et al., 2023; Hu et al., 2025), but may sacrifice interpretability and intervention efficacy. Discrete activations support symbolic reasoning, explicit rule-based interventions, and logical consistency (Dominici et al., 2025; Stammer et al., 2024; van Krieken et al., 2025), but introduce challenges in learning and may be too rigid for nuanced concepts. Distributional representations model uncertainty in concept presence or meaning, which is critical under noisy annotations, weak supervision, or distribution shift. Approaches include stochastic concept models (Vandenhirtz et al., 2024; Marconato et al., 2022; Kim et al., 2023a) and energy-based formulations (Xu et al., 2024; Kim et al., 2024) that provide unified probabilistic frameworks for joint concept reasoning and intervention. While such representations better capture uncertainty, they add computational complexity. Overall, no single representation has emerged as a universal solution. Rather, the choice depends on the available supervision and the concrete task requirements, e.g., expressiveness, interpretability, or robustness.

**Concept Absence.** A key question in the design of the concept module is how to represent the absence of a concept. In many CBMs, absence is modeled implicitly by zero-valued continuous or discrete concept activations, which conflates three semantically distinct states: weak evidence for presence, explicit negation (evidence *against* presence), and unknown status. For example, a low activation of the concept "outdoors" may indicate that the scene is unlikely to be outdoors, definitely not outdoors, or that this information is unavailable. Without a way to represent when concepts are absent, CBMs cannot provide contrastive explanations such as "it is a living room because there is a couch AND it is not outdoors.". While such phenomena could also be modeled with epistemic uncertainty estimations, adding a notion of concept absence directly to the concept representation would improve the model's ability to capture the distinction between absence, negation, and uncertainty. Moreover, interventions that set a concept activation to zero become ambiguous across these states, complicating consistent interventions (Shin et al., 2023). While concept embedding models (CEMs) (Espinosa Zarlenga et al., 2022) explore the modeling of negated concepts via an explicit concept embedding generator for concept absence, this issue remains largely unexplored despite its implications for interpretability and reliability in CBMs.

**Grounding, Concept Leakage and Input Alignment.** A key aspect of the concept module is *grounding*: ensuring concept activations align with the actual presence of concepts in inputs rather than spurious correlations (Margeloiu et al., 2021). Although many works focus on matching concept activations to labeled samples, fewer examine whether activations are genuinely grounded in the intended concepts (Havasi et al.,

2022). Joint or sequential training of the training module can result in *concept leakage*, where activations encode more information than intended, undermining interpretability and intervenability (Havasi et al., 2022). In terms of the Markov assumption ($y \perp x \mid c$), leakage indicates that the bottleneck fails to enforce conditional independence: concepts encode additional, unintended input signals that compromise interpretability. The choice of representation affects leakage susceptibility, with continuous representations more prone to leakage than discrete ones (Havasi et al., 2022). A closely related but distinct failure mode, studied in neurosymbolic predictors, is that of reasoning shortcuts: concepts that satisfy the training objective while encoding unintended semantics, even when the predictor is a structured logic or program module (Marconato et al., 2025; Bortolotti et al., 2025) To address leakage more broadly, Marconato et al. (2022) introduce prototype representations to detect when sample representations deviate excessively from prototypes, Kalampalikis et al. (2025) use an explicit side-channel for information not captured in concepts to control information flows and Almudévar et al. (2025) incorporate an additional training objective controlling the information bottleneck. Furthermore, several works address whether concepts localize to appropriate input regions through spatially structured representations based on pixels (Stropeni et al., 2025), patches (Panousis et al., 2024), or objects (Srivastava et al., 2024; Steinmann et al., 2025; Eisenberg et al., 2025). Such structures facilitate input alignment by linking concepts to explicit regions. Measuring leakage and grounding quality remains challenging, with approaches including information-theoretic metrics (Makonnen et al., 2025), intervention studies (Shin et al., 2023), and prototype-based detection (Marconato et al., 2022), although no standard evaluation protocol has emerged. Thus, achieving successful grounding remains a key challenge for CBMs.

**Completeness.** Next to understanding how a concept should be represented, a central question is which concepts should be considered in the first place. Completeness refers to whether a concept set forms a sufficient statistic for predicting the target, i.e., whether all task-relevant information is captured by the bottleneck (Feng et al., 2025; Liu et al., 2025; Tan et al., 2024a). In practice, the complete set of relevant concepts is rarely known a priori, and assessing completeness is difficult without ground-truth knowledge of which concepts are necessary. Nevertheless, considering completeness is particularly important when concept annotations are unavailable, as a complete concept set is a necessary condition for solving a task, and improving completeness can reduce the interpretability-accuracy gap. Furthermore, lacking completeness can lead to concept leakage: when task-relevant information lacks designated concepts, models may encode it implicitly through distributed activations or correlations with other concepts (Havasi et al., 2022). At the same time, excessively large *overcomplete* concept sets often include redundant representations, thereby hindering interpretability (Yan et al., 2023). Thus, several works introduce approaches to obtain better completeness: Liu et al. (2025) propose a hybrid architecture that combines static, human-defined concepts with a dynamic concept bank of unsupervised latent concepts to recover missing information from the static set. Shang et al. (2024) and Bhan et al. (2025) incrementally build the concept set based on a number of candidates given their task utility. Other works leverage vision-language models to generate comprehensive concept vocabularies (Oikarinen et al., 2023; Yang et al., 2023), while unsupervised approaches discover concepts bottom-up to avoid a priori incompleteness (Stammer et al., 2024). Some works accept potential incompleteness by incorporating residual connections or mixed architectures that allow information to bypass the concept layer (Ismail et al., 2024; Sawada & Nakamura, 2022). Overall, balancing completeness and interpretability remains a central challenge in concept semantics.

**Sparsity.** While completeness addresses the performance of a CBM, *sparsity* of its concept activations and predictor weights is critical to maintain interpretability. Predictions that rely on many concepts simultaneously are difficult for humans to interpret, making sparsity increasingly important as the concept set grows, particularly in overcomplete settings. Several approaches introduce sparsity in the concept activations, either via a human-in-the-loop (Delfosse et al., 2024), by optimizing for similar but sparse concept activations (Bhalla et al., 2024), or by adding a regularized, input-dependent concept selector (Schrodi et al., 2025). Other approaches argue that sparsity should be induced in the predictor (output module) rather than the concept activations (concept module). Turan et al. (2025) achieve this by training the predictor with a prover-verifier game that incentivizes sparse predictions, while Du et al. (2025) introduce a sparsemax module that enforces sparsity in the predictor's weights. While different approaches to sparsity exist, the advantages of activation-level vs. predictor-level sparsity remain unclear, and systematic comparisons are limited. Nevertheless, sparsity is essential to maintain the interpretability of CBMs.

**Assuming Concept Annotations.** Early CBM formulations often assumed access to complete and detailed concept-level annotations (Koh et al., 2020; Espinosa Zarlenga et al., 2022; Losch et al., 2021; Stammer et al., 2021). This enabled these models to match or even outperform black-box baselines and provided valuable guidance and alignment signals when available. However, such comprehensive annotations represent a practical burden as they require extensive expert effort to produce and, in many cases, a complete concept set may not even be known in advance. More recent work has therefore sought to relax this assumption through different strategies. One line of work leverages pretrained vision-language encoders such as CLIP (Radford et al., 2021), relying on their learned capabilities for concept detection (Yang et al., 2023; Oikarinen et al., 2023; He et al., 2025c), though this may simply trade one limiting assumption (concept annotations) for another (access to aligned foundation models) (Debole et al., 2025). Alternatively, unsupervised methods based on disentanglement (Rao et al., 2024; Yuksekgonul et al., 2023) or dictionary learning (Stammer et al., 2024; Lee et al., 2025b) discover concepts directly from data without annotations. However, these approaches face distinct challenges: discovered concept spaces are often high-dimensional (e.g., hundreds or thousands of features), making human inspection potentially difficult. A further alternative comes from neurosymbolic approaches, where structured background knowledge, expressed for instance as logical rules, programs, probabilistic constraints, or differentiable reasoning modules, can provide weak supervision during concept learning (Skryagin et al., 2023; Marconato et al., 2023b; 2025). Rather than requiring exhaustive concept labels, such constraints restrict which concept configurations are consistent with the task, implicitly guiding concept discovery. This trades annotation effort for the availability of domain knowledge, offering a complementary pathway in settings where symbolic background knowledge exists but fine-grained instance-level annotations do not. Importantly, the choice of concept supervision fundamentally shapes the entire concept module: it determines whether semantics or grounding comes first (top-down vs. bottom-up), influences the granularity and interpretability of concept representations, and can even dictate the form of concept activations (e.g., whether concepts are discrete, continuous, or distributional). Overall, the tension between annotation requirements, model expressiveness, and true interpretability remains an open challenge.

## 5.2 Model Reliability

Many works focus primarily on improving the architecture of CBMs to increase predictive performance. In contrast, the following challenges concern broader questions about the behavior and reliability of CBMs.

**Robustness.** CBMs are not inherently less robust than other model types. However, robustness issues can arise in the input encoder, the concept source, or the predictor (i.e., spanning components of the input, concept and output module), degrading both predictive performance and the reliability of concept-based explanations (Raman et al., 2023). Potential causes of a lack of robustness include sensitivity to adversarial perturbations or input noise, brittleness under noisy or incomplete concept annotations, and performance degradation under distribution shifts between training and deployment. As challenges like adversarial robustness and input noise sensitivity are largely inherited from the used backbone modules, CBM-specific robustness research has focused on challenges more directly tied to the concept module itself. To improve robustness to distribution shifts, Choi et al. (2025) include a detector for shifts at test time and respond by dynamically adapting the linear classifier and the concept set. Park et al. (2025) address robustness to noisy concept annotations through sharpness-aware minimization to stabilize training and uncertainty-based intervention strategies to correct low-confidence concept predictions at inference. Similarly, Kim et al. (2024) utilize probabilistic concept activations to capture concept uncertainty to improve robustness. Although these approaches address specific robustness issues, there is still no unified framework that incorporates robustness across input perturbations, annotation quality, and distribution shifts.

**Spurious Correlations and Reasoning Shortcuts.** One reason for unreliable behavior in models can be spurious correlations (Ross et al., 2017; Schramowski et al., 2020; Geirhos et al., 2020; Steinmann et al., 2024a). For CBMs, such spurious correlations can appear in the input encoder and predictor. Specifically, the input encoder may learn to incorporate spurious features within the input that correlate with concepts but do not reflect their intended semantics. For example, a model might associate the appearance of "grass" with the concept of "outdoors" as outdoor scenes often include grass. On the other hand, the predictor may also exploit spurious correlations between latent concepts and prediction labels (Delfosse et al., 2024). Recent

work has begun to expose such failure modes, showing that CBMs can still rely on non-causal cues despite an interpretable interface (Bortolotti et al., 2025), but systematic mitigation strategies remain limited. A common approach to identify and mitigate such issues is to incorporate iterative human-in-the-loop feedback, e.g., via corrective losses (Stammer et al., 2021), interventions (Steinmann et al., 2024b; Bhalla et al., 2024), or other weak (Delfosse et al., 2024) or data-centric supervision (Stammer et al., 2022).

**Uncertainty.** Quantifying uncertainty in the concept activations and the final predictions is an essential step towards reliable and faithful CBMs. Internal uncertainty measurements can be important to improve model robustness, serve as a central mechanism to enable targeted interventions and selective deferral (Shin et al., 2023; Park et al., 2025; Pugnana et al., 2025). Importantly, concept-level uncertainty provides interpretable signals about where a model is uncertain, not just that it is uncertain, making CBMs particularly well-suited for uncertainty-aware decision-making. Existing approaches operationalize uncertainty at different levels of the CBM pipeline. Zhang et al. (2025) measure uncertainty via the Euclidean distance between an instance's concept representation and a class-specific concept prototype. Beyond point estimates, probabilistic CBMs (Kim et al., 2023a) explicitly model each concept as a distribution by learning concept-specific uncertainty parameters; concept activations are obtained by sampling from these distributions and comparing samples against learned positive and negative anchors. More recently, van Krieken et al. (2025) replace the common independence assumption between concepts by using discrete masked diffusion to model dependencies among concepts, enabling uncertainty-aware reasoning. Although these methods introduce different mechanisms for quantifying uncertainty in CBMs, their calibration properties, computational trade-offs, and relative effectiveness across different scenarios have yet to be systematically evaluated.

**Interventions.** An additional strength of CBMs lies in the ability of users to interact directly with the concept activations. In particular, interventions can provide a practical mechanism for model debugging, causal analysis, and human-in-the-loop interaction (Shin et al., 2023; Steinmann et al., 2024b; Laguna et al., 2024), which are essential tools for analyzing and improving a model's reliability. Consequently, a line of work has been carried out analyzing and improving the intervention procedures for CBMs. Shin et al. (2023) explore various intervention strategies and evaluate how aspects such as concept selection, ordering, or correction magnitude affect downstream prediction and faithfulness. To avoid the necessity to check each individual concept, Chauhan et al. (2023) propose a method to automatically select concepts that might need human interaction. When individual concepts have been corrected, Singhi et al. (2024) introduce a learnable module that automatically updates further concepts based on individual interventions. Vandenhirtz et al. (2024) achieve a similar effect by explicitly modeling correlations between concepts and propagating interventions to multiple concepts accordingly. To improve the efficiency of interventions on multiple samples, Steinmann et al. (2024b) introduce a learnable memory that memorizes interventions on previous samples and automatically reapplies them to other samples if applicable, while also suggesting samples that should be checked by a user. Going beyond interventions at individual concept levels, He et al. (2025b) utilize a frozen LLM to enable conversational interventions with reasoning guidance. To bring the advantage of interventions to non-CBMs, Laguna et al. (2024) even explore the introduction of an implicit concept-layer within black-box models. Although recent work has improved efficiency and practical usability, systematic evaluation with real users is still scarce. Open questions remain how interventions should propagate across concepts, how to balance automation with user control, and how to assess intervention faithfulness.

**Causality.** Despite their motivation for interpretable reasoning, most CBMs rely on correlational concept predictors and lack explicit modeling of causal relationships between concepts and labels. As a result, interventions on concept activations do not necessarily correspond to valid *causal* manipulations, limiting the reliability of counterfactual reasoning. Key challenges for this include: (i) causal structure is difficult to identify from observational data alone; (ii) concepts often exhibit causal dependencies among themselves that most CBMs ignore; and (iii) concept sets must include confounders and mediators to support valid causal interventions, a stronger requirement than information-theoretic completeness. De Felice et al. (2025) target these issues by using LLM-based retrieval for concept and causal graph discovery, then train a neural encoder coupled with a structural causal model reflecting the discovered graph structure. Asiyabi et al. (2026) also include external information about causal relationships, specifically known dependencies between biophysical

concepts for biomass density estimation. Despite these initial steps, establishing proper causal foundations for CBMs remains an important open direction.

### 5.3 Validation and Scope

This section explicitly covers challenges related to the quality and application of CBMs.

**Performance Evaluation and Benchmarking.** An important challenge for the development of CBMs is the question of how they should be evaluated. The absence of standardized metrics that jointly capture predictive performance, different angles of interpretability, and utility for intervention makes this a difficult task. Existing evaluations focus mostly on task accuracy, concept prediction quality, or intervenability, but rarely assess their interplay. Beyond task accuracy, evaluating whether concepts are genuinely grounded in the intended input patterns rather than leaking unintended information remains an open problem, with approaches ranging from information-theoretic metrics (Makonnen et al., 2025) to intervention studies and prototype-based detection, but no standard protocol has emerged. Similarly, while interventions are a central property of CBMs, how to systematically evaluate intervention procedures, including their faithfulness and real-world utility with human users, remains largely unresolved (Shin et al., 2023). The amount and quality of benchmark datasets further complicate the evaluation: widely used datasets such as CUB (Wah et al., 2011) exhibit annotation biases and limited complexity, while many benchmarks without concept annotations lack structured methods to evaluate concept representations. Moreover, benchmarks are heavily concentrated on image classification, with evaluation resources for text, video, time-series, tabular, and generative CBMs remaining scarce, making it difficult to assess how CBM approaches generalize across modalities and task types. To address some of these issues, Bader et al. (2025) introduce a controlled benchmark of synthetic images based on CUB that is based on a subset of classes and concepts and contains explicit variations of both, so that CBM robustness can be evaluated in a controlled, but large-scale setting. Despite such initial efforts, a CBM-specific benchmark suite that explicitly evaluates the quality, grounding, and use of concepts across diverse modalities and tasks, rather than accuracy alone, remains missing and is essential for systematically investigating the trade-offs among these aspects.

**Evaluating Interpretability.** Although interpretability is a defining motivation of CBMs, evaluating it is arguably more challenging than evaluating predictive performance, as the underlying notion lacks a commonly accepted definition. Different works operationalize interpretability in different, often implicit ways, ranging from concept consistency to the interventions effectiveness and the sparsity or inspectability of explanations. Some works introduce interpretability metrics such as concept consistency, grounding quality, and leakage detection (Parchami-Araghi et al., 2025; Bader et al., 2025; Marconato et al., 2022), which measure technical properties of the bottleneck rather than, e.g., user understanding. User studies (Wang et al., 2023; Rao et al., 2024; Yuksekgonul et al., 2023), though potentially more direct, remain scarce in the CBM literature. When conducted, they often focus on whether concept names are meaningful to users rather than whether users can effectively leverage the model's explanations for decision-making or debugging. Such studies are often subjective, difficult to reproduce, and can miss the rigor expected of human interaction studies. While some works cite model debugging via explanations as evidence of utility (Stammer et al., 2021; Delfosse et al., 2024; He et al., 2025a), this remains an indirect proxy and is often simulated, rather than evaluated via human users. Thus, systematic evaluations of whether practitioners can use explanations for debugging and whether this improves outcomes remain rare and interpretability assessments remain largely fragmented.

An additional tension exists between interpretability and expressivity. Sparse activations arguably enhance interpretability by limiting active concepts per prediction, whereas larger, densely activated concept sets often improve performance (Srivastava et al., 2024). Yet many works report results without accounting for concept set size, implicitly favoring larger sets and obscuring this trade off (Yan et al., 2023). This bears the consequence: focusing primarily on accuracy can lead to models that diverge from their intended reasoning process, revealing failures in concept usage and faithfulness despite strong predictive performance (Baniecki & Biecek, 2025; Ramaswamy et al., 2023; Almudévar et al., 2025). Resolving these issues requires not only additional evaluation metrics that comprehensively measure CBM performance beyond accuracy alone (Espinosa Zarlenga et al., 2023a; Parchami-Araghi et al., 2025), but also a shared conceptual foundation on what interpretability in CBMs should actually mean.

**Broader Tasks.** CBMs were initially introduced for image classification tasks (Koh et al., 2020; Stammer et al., 2021), and most research in the field has continued to focus on this domain, particularly on standard benchmark datasets. While image classification remains a natural starting point due to the ease of identifying visual concepts even as a lay user, CBMs have been extended to diverse domains, modalities, and task types. We organize these extensions by increasing departure from the original formulation, noting that each modifies one or more of the four modules defined in our taxonomy.

*Specialized Domains Within Vision.* Several works have applied CBMs to specialized image domains requiring domain-specific concept sets. In medical imaging, CBMs have been used to identify diseases from skin lesions (Chowdhury et al., 2024) and chest x-rays (Mpinda et al., 2026), where concepts correspond to clinical features. Asiyabi et al. (2026) predict biophysical concepts from satellite images for above-ground biomass density estimation, demonstrating applicability to remote sensing.

*Extended Modalities and Data Types.* Moving beyond static images, several works have extended CBMs to modalities with temporal, sequential, or structured characteristics. Video-based CBMs learn motion-aware concept representations that capture dynamic patterns: Lee et al. (2025b) explore explainable video action recognition through concepts spanning discrete prototypes and continuous embeddings, Knab et al. (2025b) introduce a channel-preserving transformer-based architecture for video understanding, and Lee et al. (2025c) replace pixel-level inputs with human-pose sequences to obtain motion-centric concepts for pose estimation. Text-based CBMs adapt concept bottlenecks to language understanding and generation: Tan et al. (2024b) leverage LLM-generated concepts with robust training under label noise for text classification, Sun et al. (2025) extend CBMs to text generation through hybrid architectures addressing concept completeness, and He et al. (2025b) employ frozen LLMs as symbolic predictors enabling conversational interventions. Beyond vision and language, Wu et al. (2022) adapt CBMs to time series by computing predefined statistics and learning to combine them into concepts, Espinosa Zarlenga et al. (2023c) apply CBMs to tabular data by defining concepts as subsets of correlated features, and Ismail et al. (2025b) employ generative CBMs for protein design. These extensions demonstrate that the concept bottleneck paradigm generalizes across modalities, though each requires rethinking what constitutes a meaningful concept in that domain.

*Extended Task Types and Multi-Task Models.* Beyond discriminative classification, CBMs have been adapted to fundamentally different task structures. Ismail et al. (2024) introduce concept bottlenecks for generative tasks by conditioning generation on an explicit concept layer while maintaining a residual pathway to preserve generative expressivity. Delfosse et al. (2024) extend CBMs to reinforcement learning, introducing multi-level concept bottlenecks for interpretable action prediction in sequential decision-making tasks. Lorello et al. (2024) adapt CBMs to continual learning settings where concepts are discovered and refined over a stream of tasks, requiring mechanisms to maintain concept stability over time. Notably, Wittenmayer et al. (2026) move beyond single-task specialization with a language-aligned concept foundation model that extracts spatially grounded, hierarchical concepts at multiple granularities, enabling interpretable decision-making across diverse vision tasks including classification, segmentation, and captioning.

Together, these works demonstrate that CBMs extend well beyond their original image classification setting. The concept bottleneck serves as a general-purpose mechanism for introducing interpretable intermediate representations across domains, modalities, and task types. However, each extension also introduces domain-specific challenges, and understanding which aspects of the concept bottleneck paradigm are universal and which require task-specific adaptation remains an important question when adapting CBMs to new settings.

**Summary.** Overall, these challenges highlight that progress in CBMs cannot be driven by architectural innovations alone. Addressing issues of reliability, faithfulness, evaluation, and semantic validity requires rethinking how concepts are defined, measured, and interacted with across the full CBM pipeline. In the following section, we outline key directions for future work that build on these open challenges.

# 6 Future Directions and Related Research

While the previous section highlighted concrete challenges in CBMs tied to aspects of the individual taxonomy components, we now turn to broader questions about the field's future direction. These go beyond individual limitations and instead focus on how CBMs may expand to new problem settings and connect to neighboring

research areas. We begin by outlining potential future research directions in Sec. 6.1, and position CBMs within a broader research context in Sec. 6.2.

## 6.1 Future Research Opportunities

**Expanding CBMs: from new modalities to tasks.** An essential consideration remains to extend CBMs beyond standard image classification toward settings that introduce fundamentally new challenges. We organize this direction along four dimensions of complexity that stress different components of the CBM taxonomy and reveal where current architectures fall short.

*Modal and Temporal Complexity.* Extending CBMs to video, time series, or multimodal inputs requires rethinking both concept representation and grounding. Many of these challenges, e.g., representing time, modality, and structured knowledge, have long been studied in the field of knowledge representation (Davis et al., 1993), which offers a source of inspiration that current CBM work has largely left untapped. In video, concepts must capture not only visual attributes but also temporal dynamics (Lee et al., 2025b). E.g., actions unfold over time, objects persist and transform, and causal relationships emerge across frames (Knab et al., 2025b). This requires *temporal* concept representations that can model concept evolution, multi-timescale reasoning (frame-level vs. event-level concepts), and temporal binding mechanisms that correctly associate concepts with specific time windows. For time series data, concepts need to capture abstract and data-dependent patterns (e.g., "rising trend", "periodic oscillation") that lack clear visual analogs. Multimodal settings introduce cross-modal grounding problems: how do we ensure concepts align meaningfully across vision, language, and other modalities? Many CLIP-based (Radford et al., 2021) CBMs make use of joint embedding spaces, but questions remain about whether concepts should be modality-specific, shared, or hierarchically organized between modalities. Progress on these fronts will also require task- and domain-specific benchmarks to establish comparable baselines across modalities.

*Compositional and Hierarchical Complexity.* Many real-world tasks require reasoning about compositional structure, which standard flat CBMs struggle with: scenes contain objects with parts, actions consist of sub-actions, and high-level concepts emerge from combinations of lower-level ones. For instance, recognizing "playing tennis" requires grounding concepts like "person", "racket", "ball", and their spatial-temporal relationships. This suggests that CBMs require structured representations, such as graph-based embeddings to capture relations (Barbiero et al., 2024; Xu et al., 2025), hierarchical concept representations where high-level concepts depend on low-level ones (Panousis et al., 2024; Wittenmayer et al., 2026), or object-centric representations where *concept groups* respect visual object boundaries (Hong et al., 2024; Steinmann et al., 2025). However, structured representations can face combinatorial scaling issues: concept spaces grow exponentially with the number of objects, relations, or hierarchical levels, challenging both interpretability (too many concepts to inspect) and computational tractability. Beyond representation, compositional reasoning also requires predictors that go beyond linear models. Logic-based modules (Debot et al., 2024; Vemuri et al., 2025) and *program-based* reasoning offer compositional and generalizable predictions, yet integrating differentiable or searchable program synthesis (Gulwani et al., 2017; Grand et al., 2024; Hsu et al., 2023; Wüst et al., 2025) with gradient-based concept bottlenecks remains underexplored, in part due to the exponentially large program search spaces involved. Addressing compositional complexity while maintaining interpretability and tractability remains a key open problem for CBMs.

*Scale and Dynamic Complexity.* As concept vocabularies grow, whether through LLM-generated concept sets (Oikarinen et al., 2023; Yang et al., 2023) or attempts to build general, comprehensive concept libraries (Ismail et al., 2025a), CBMs face scalability challenges. Large static concept sets become computationally expensive and may include many irrelevant concepts for any given input. This motivates dynamic mechanisms (Chowdhury et al., 2024) that selectively activate and reason over task-relevant concept subsets. Importantly, in such scenarios, the predictor must also adapt: simple linear layers become inadequate for reasoning over hundreds or thousands of concepts, while complex predictors sacrifice interpretability. Recent work explores symbolic reasoning modules (Vemuri et al., 2025; Debot et al., 2024), LLM-based reasoning (He et al., 2025b; Patrício et al., 2025a), and multi-agent learning approaches (Turan et al., 2025) as viable alternatives, though balancing expressiveness with transparency remains an open challenge.

*Generative Complexity.* The rapid adoption of generative models, from diffusion models to large language models, introduces perhaps the most fundamental challenge: *What constitutes a concept-based explanation for generation?* In discriminative settings, concepts explain "Why this prediction?", but in generative settings, the target is unclear. Should explanations describe the latent space structure, the generation trajectory through denoising steps, or the semantic attributes of generated outputs? Integrating concept bottlenecks into generative architectures (Ismail et al., 2025b; Kulkarni et al., 2025b) also raises questions about semantic consistency (do interventions on concepts produce semantically coherent outputs?) and completeness (can the concept space capture all generative factors?). Recent work on diffusion model interpretability (Gandikota et al., 2023; Kumari et al., 2023) and LLM interpretability offers some insights (Bricken et al., 2023; Huben et al., 2024): concepts can be identified post-hoc in latent spaces, where directions can correspond to interpretable transformations, and concept-guided generation can potentially be achieved through steering (Arad et al., 2025; Härle et al., 2025). However, these approaches often bypass the bottleneck structure central to CBMs. A fundamental open question remains: can generative CBMs provide the same interpretability guarantees as discriminative ones, or does generation require richer, less constrained representations?

**Symbol Grounding and the Binding Problem.** CBMs address two fundamental challenges in AI and cognitive science: the symbol grounding problem and the binding problem. The symbol grounding problem asks how symbols acquire meaning beyond circular definitions, i.e., how they connect to what they represent in the world rather than just to other symbols (Harnad, 1990). The binding problem concerns how separate features in different parts of a system are combined into coherent and unified representations (Treisman, 1999; Feldman, 2013; Greff et al., 2020). CBMs have connections to both: they attempt to ground high-level (symbolic) concepts (interpretable labels like "contains a TV" or "is indoor") in perceptual features, while the bottleneck architecture explicitly structures which concepts bind to which inputs, forcing the model to make feature-to-concept associations explicit rather than leaving them entangled in distributed representations.

However, it is unclear to what extent current CBMs achieve genuine semantic grounding and robust binding. While concepts are anchored in training data or pretrained representations, this grounding often amounts to labeling rather than understanding. Similarly, common issues in CBMs (cf. Sec. 5) such as concept leakage (where information bypasses the bottleneck), poor localization (where concepts fail to capture spatially coherent features), and binding errors (where concepts misattribute features) reflect the underlying difficulty of the binding problem itself. These are not merely implementation flaws of CBMs, but manifestations of fundamental challenges that persist across AI architectures. Indeed, standard neural networks exhibit analogous problems: feature binding errors in vision models (Pantazopoulos & Özyiğit, 2025), difficulties with compositional generalization in foundation models (Li et al., 2025; Bayat et al., 2025; Woydt et al., 2025), and the challenge of learning object-centric representations in scene understanding (Li et al., 2020; Didolkar et al., 2024; Heo et al., 2025). Rather than viewing issues such as concept leakage, poor localization, or incomplete binding as architectural failures unique to CBMs, they are manifestations of the fundamental challenge of connecting symbolic and perceptual representations. Recognizing this framing implies that CBM limitations should be contextualized within broader unsolved problems and that solutions from other communities addressing grounding and binding may transfer to CBMs. Recent neurosymbolic work, for instance, formalizes these challenges through the lens of reasoning shortcuts and symbol alignment (Marconato et al., 2025), offering diagnostic frameworks that distinguish whether grounding failures stem from concept representations, the learning process, or misalignment between task structure and concept vocabulary.

**Generalist vs. Specialist Models.** As discussed above, the emergence of large-scale foundation models has fundamentally shifted the landscape of concept-based learning. Traditionally, CBMs were trained on specific tasks with predefined concept vocabularies, requiring careful domain expertise and labeled concept annotations. In contrast, modern foundation models pretrained on large and diverse datasets provide rich, general-purpose representations that support a wide range of downstream tasks. This raises critical questions about the path forward for concept-based models: should these aim to produce large, generalist models with expansive concept vocabularies (Ismail et al., 2025a; Wittenmayer et al., 2026), remain specialized models optimized for task-specific interpretability, or adopt hybrid approaches that dynamically extract task-relevant concept sets from a generalist backbone (Chowdhury et al., 2024)? More broadly, it remains unclear whether

concept learning is best achieved by training specialist models from scratch or by leveraging the semantic knowledge embedded in pretrained foundation models such as CLIP (Oikarinen et al., 2023).

Generalist models offer advantages such as zero- or few-shot concept recognition and reduced annotation effort, but they may inherit biases from web-scale training data and introduce substantial computational overhead. More fundamentally, a well-trained concept layer could itself enable a form of zero-shot generalization: novel classes might be specified simply by defining a new concept-to-class mapping rather than collecting labeled training data (Prasse et al., 2025). Whether CBMs can reliably support this weaker but practically relevant form of transfer, and how robust it is under concept distribution shift or incomplete concept coverage, remains an open question. In contrast, specialist models can be carefully tailored to specific use cases, but often struggle to generalize beyond them and can suffer from issues such as limited data of these use cases. Overall, the trade-offs between generalist and specialist CBMs remain underexplored. In particular, systematic studies are needed to assess both approaches in more practical environments, such as the balance between annotation cost for specialized models and potential losses in concept quality when relying on pretrained generalist representations (Debole et al., 2025).

**Toward Dynamic, On-Demand Concept Bottlenecks.** It is also worth discussing that the "bottleneck" of CBMs is often viewed as a permanent architectural trade-off, trading flexibility and performance for inherent interpretability. However, this framing couples the potential price of interpretability with every use of the model, although it is not always required. A more flexible view treats concept bottlenecks as dynamic, on-demand sub-models (Shang et al., 2024; Chowdhury et al., 2024), where an opaque model operates by default, and the interpretable bottleneck is activated only when interpretability or accountability is required. This relates naturally to the distinction between System 1 and System 2 reasoning (Kahneman, 2011), in the sense that most of the time, fast, but opaque processing might suffice, but a corresponding concept bottleneck can be used when deliberate, inspectable reasoning is required.

A critical property for this approach is *faithfulness*: the bottleneck must remain grounded in the opaque model's internals. Concretely, the bottleneck should be understood as the best possible simplification of the original model's representations given the expressiveness constraints of the concept space. This grounded relationship can be established either jointly during training or post-hoc, provided the bottleneck remains tied to the underlying model's representations (Schrodi et al., 2025; Stammer et al., 2024). More broadly, the idea of selectively routing between opaque and interpretable components has been explored in the hybrid predictive modeling literature (Wang & Lin, 2021; Ferry et al., 2024; Frost et al., 2024), where an interpretable model handles a subset of instances and defers to a black-box model otherwise, often gaining transparency at little or no cost to accuracy. Extending this paradigm to concept bottleneck models where a faithful, concept-based sub-model is activated on demand while remaining grounded in the underlying model's representations is a promising future direction.

**Agentic AI and Concept Bottleneck Models.** The emerging paradigm of agentic AI, i.e., systems capable of autonomous, goal-directed behavior over extended sequences, presents potential opportunities for interpretable AI overall (Kim et al., 2025), but also concept-based models. While this intersection remains largely unexplored, the relationship could be bidirectional: agentic methods might improve CBM development, while concept-based architectures might provide interpretable substrates for agentic reasoning.

*Potential for Agentic CBM Construction.* Agentic systems have the potential to address a fundamental bottleneck in CBM development: the substantial manual effort required to design, label, and validate concept sets. Rather than relying solely on static, human-defined concepts, agentic approaches might autonomously explore datasets to identify missing or redundant concepts (Shaham et al., 2024), strategically query domain experts through active learning, or search scientific literature and ontologies to refine concept definitions. Multi-agent systems could potentially divide responsibilities across concept discovery, grounding validation, and deployment monitoring, enabling self-improving CBMs that detect failures and propose refinements. Thus, while current CBM development remains largely manual, agentic capabilities could substantially reduce expert effort, particularly in specialized domains.

*Potential for Concept-Based Agentic Systems.* Conversely, CBMs might serve as interpretable reasoning substrates for agentic systems. Concept-based agents could plan over higher-level abstractions rather than raw

observations, potentially improving sample efficiency and enabling hierarchical planning while maintaining transparency. Perhaps most importantly, concept bottlenecks might enable human oversight: humans could monitor an agent's concept-level reasoning and intervene when behavior appears misaligned, offering a more principled approach to human-agent collaboration. Notably, such concept-level monitoring and intervention mechanisms are already being explored in Vision–Language–Action (VLA) systems for safety-critical settings, where structured intermediate representations are used to detect, constrain, or override unsafe behaviors (Zhou et al., 2025; Shi et al., 2025).

## 6.2 Broader Relation to Other Fields

CBMs are not always clearly distinguishable from other fields, and in the course of reviewing existing work we noticed that several other communities share common ideas and goals with concept-based learning. Some works do not explicitly frame themselves as CBMs but employ similar core mechanisms, i.e., extracting interpretable intermediate representations and reasoning over them. We set CBMs in context of these broader related fields below, highlighting both the overlaps and the distinctions.

*Interpretable Representation Learning & Mechanistic Interpretability.* A closely related line of work concerns learning interpretable latent representations without an explicit task-driven bottleneck. Variational Autoencoders (VAEs) (Kingma & Welling, 2014; Higgins et al., 2017) and variants (van den Oord et al., 2017) aim to learn compressed latent spaces in which individual dimensions ideally correspond to meaningful, independent factors of variation. This objective aligns closely with those of concept learning in the context of CBMs, though they typically work without explicit human semantic alignment. More recently, Sparse Autoencoders (SAEs) have gained significant traction in the mechanistic interpretability community (Bricken et al., 2023; Hindupur et al., 2025; Huben et al., 2024), where they are used to decompose neural network activations into sparse, interpretable features. Both SAEs and CBMs seek to extract meaningful features from opaque activations and both aim to make model internals transparent. The key distinction lies in supervision and structure: CBMs impose explicit semantic constraints (e.g., concept names, human-defined vocabularies), while SAE-based approaches discover features in an unsupervised manner. This suggests these represent complementary rather than competing approaches, e.g., SAEs for discovery, CBMs for structured, explainable reasoning. Recent work has made this connection more formal: Rocchi-Henry et al. (2025) show that CBMs and SAEs instantiate the same underlying geometric structure, each learns a set of linear directions whose nonnegative combinations form a "concept cone", and differ only in how that cone is selected (supervision vs. sparsity). In fact, recent work has begun employing SAEs directly for concept bottleneck construction (Rao et al., 2024; Schrodi et al., 2025; Kulkarni et al., 2025a) and other forms of structured reasoning (Helff et al., 2026). Relatedly, circuit discovery methods from the mechanistic interpretability literature (Olah et al., 2020; Conmy et al., 2023; Marks et al., 2025) aim to identify the specific sub-networks (circuits) within a model that implement particular functions or represent particular features. While CBMs characterize *what* a model represents in terms of human-understandable concepts, circuit discovery characterizes *how* these representations are computed. These perspectives can be seen as complementary, e.g., use circuit-level analysis for building faithful concept bottlenecks.

*Causal representation learning.* Causal representation learning (CRL) aims to discover latent variables that correspond to underlying causal factors and their graph directly from data (Schölkopf et al., 2021). In an ideal case where CBM concepts are chosen to align with true causal factors and structured according to their causal graph, a CBM can be seen as an instance of CRL with an interpretable, supervised causal bottleneck (Marconato et al., 2023a). Recent work formalizes this connection by structuring the concept layer as a causal graph to support interventions and improve robustness to spurious correlations De Felice et al. (2025), or by assuming human-defined concepts correspond to causally identified variables Fokkema et al. (2025). However, the connection between causal representation learning and concept bottleneck models is fragile: only when concepts are carefully aligned with underlying causal mechanisms and implemented via causally structured, shortcut-resistant architectures do such bottlenecks provide causal, rather than correlational, benefits.

*Neuro-Symbolic AI.* Neuro-symbolic AI shares with CBMs the goal of making learned representations explicit and structured, but operates with broader ambitions. While CBMs focus primarily on interpretability and intervention, neuro-symbolic systems aim additionally for greater robustness, systematic generalizabil-

ity, and formal reasoning capabilities (Yi et al., 2018; Shindo et al., 2023; Barbiero et al., 2023; Garcez & Lamb, 2023; Jones, 2025). The overlap is particularly clear in systems that decompose perception and reasoning: a neural network extracts symbolic primitives from raw input, and a symbolic reasoner operates over them. Under this view, CBMs can be seen as a constrained instance of this paradigm: the concept bottleneck is the symbolic interface, and the predictor is the reasoning module. However, neuro-symbolic systems can go further by incorporating logical rules (Shindo et al., 2023), program induction (Ellis et al., 2021), probabilistic reasoning (Manhaeve et al., 2018; Skryagin et al., 2023), or formal verification (Xie et al., 2022; Kouvaros, 2023). These mechanisms can enable additional formal guarantees such as consistency checking, uncertainty quantification over symbolic states, or verification against specifications, at the potential cost of more complex training and a greater risk that the learned representations do not cleanly match the symbols the reasoner expects. CBMs, by contrast, constrain the predictor to remain interpretable, trading expressiveness for transparency, though much research has gone into recovering both. Despite these differences, the boundary between the two paradigms is becoming increasingly fuzzy. Several challenges identified in CBMs, notably shortcut learning and reasoning artifacts, are also recognized in neuro-symbolic systems, pointing to shared difficulties in grounding learned representations in symbolic operations (Marconato et al., 2025). Conversely, recent CBMs that pair discrete concept representations with program-based or logic-enhanced predictors (Debot et al., 2024; Vemuri et al., 2025) directly instantiate the neuro-symbolic paradigm. As CBMs continue to move toward more compositional and structured settings, convergence with neuro-symbolic approaches is a promising direction.

*Scene Graph Generation, Segmentation, and Captioning.* In the context of image processing, several computer vision tasks produce structured intermediate representations that relate to concept bottlenecks, yet are rarely discussed in the CBM literature. Scene Graph Generation (SGG) extracts objects, their attributes, and relations from images into a structured graph, essentially representing a bottleneck with explicit relational structure (Johnson et al., 2015; Krishna et al., 2018; Shi et al., 2019; Li et al., 2024). Image segmentation frameworks decompose scenes into object masks and regions, providing spatially grounded concept-like representations (Long et al., 2015; Kirillov et al., 2023; Minaee et al., 2021; Zhou et al., 2024). Visual captioning produces natural language descriptions that implicitly define a concept set for the image (Vinyals et al., 2015; Abdulgalil & Basir, 2025). All three tasks create what could be viewed as a structured bottleneck: an interpretable intermediate representation that mediates between raw perception and downstream reasoning. The distinction is largely one of framing and objective, yet the representations produced are functionally similar. This suggests potential for tighter integration, e.g., scene graphs could extend and inform current CBM concept spaces, segmentation masks could provide spatial grounding for concepts, and captioning models could serve as concept naming mechanisms. Indeed, recent CBM work has begun exploiting such connections (Hong et al., 2024; Steinmann et al., 2025; Eisenberg et al., 2025).

*LLM Reasoning and the Perception-Abstraction-Reasoning Decomposition.* One way modern LLMs perform reasoning reveals a structural parallel to CBMs. Techniques such as token-level Chain-of-Thought prompting (CoT) and variants (Wei et al., 2022; Yao et al., 2023; Besta et al., 2024) encourage models to produce explicit intermediate reasoning steps before arriving at a final answer, effectively imposing a soft bottleneck of discrete tokens between perception (understanding the input) and reasoning (deriving the answer). Neuro-symbolic LLM reasoning approaches with formal solvers (Pan et al., 2023; Helff et al., 2026) impose a harder architectural boundary. Although it remains an important question whether CoT tokens genuinely reflect a model's reasoning process (Turpin et al., 2023), the parallel is a decomposition into perceive → abstract → reason: token-level LLM reasoning approaches make the abstraction step explicit in token space, while CBMs make it explicit in concept space. The difference lies in the degree of structure and constraint: CoT intermediate steps are unconstrained natural language, where the boundaries between perception, abstraction, and reasoning remain fluid, whereas CBM concepts are structured, named, and interventionable.

Across all these related fields, a common thread emerges: the potential of intermediate, structured representations between perception and reasoning. Understanding CBMs as one instantiation of this broader perceive-abstract-reason paradigm, rather than an isolated architecture, positions the field to draw on insights from other research communities, while clarifying what is distinctive about concept-based approaches: the explicit, human-aligned, interventionable nature of the abstraction layer.

# 7 Conclusion

This survey provided a structured and comprehensive overview of Concept Bottleneck Models, categorizing a growing and increasingly fragmented body of work into a unified taxonomy. Rather than benchmarking models on specific datasets or metrics, we organized the design space and clarified how methods relate conceptually, providing a foundation for future empirical comparison. We decomposed the CBM framework into its core modules (cf. Fig. 4) along the dimensions of *input*, *concept*, *output*, and *training* and explicitly separated concept semantics from concept grounding, thereby clarifying key design choices, underlying assumptions, and trade-offs that are often conflated in the existing literature. This perspective enabled a systematic comparison of CBM variants while highlighting recurring challenges and open questions along the four dimensions. Our taxonomy revealed how innovations and challenges systematically map onto these four modules: input module choices determine what information is preserved for concept extraction; concept module design balances expressiveness, interpretability, and human alignment; output module decisions shape intervention mechanisms and reasoning capabilities; and training strategies influence whether the Markovian assumption holds in practice. By decomposing CBMs along these dimensions, our framework makes explicit which component-level choices drive specific strengths or failure modes.

For researchers, this taxonomy serves as both a diagnostic and design tool, enabling systematic comparison between methods and revealing opportunities for targeted improvements within specific modules. For practitioners, the survey underscores that deploying CBMs in real-world settings requires careful consideration of supervision costs, intervention reliability, and the practical utility of concept-level explanations for end users. Looking forward, addressing the open challenges we identified, from compositional reasoning and dynamic concept spaces to robust grounding and evaluation protocols, will be essential for realizing the full potential of concept-based interpretability in AI.

The field of concept bottleneck models has grown rapidly in recent years, accompanied by a surge of new methods, empirical advances, and emerging challenges. As the literature expands, maintaining conceptual coherence becomes increasingly important. New approaches should clearly specify which components of the CBM architecture they modify, relate their contributions to prior work, and evaluate models in a structured and systematic manner. Crucially, CBMs should not lose sight of their defining objective: human interpretability. While many recent works focus on scaling and downstream performance, interpretability is often treated only as a side objective. As discussed in the previous sections, core challenges such as sparsity, concept set design, and representation structure are inherently tied to interpretability and must remain a central point of attention. Likewise, when incorporating powerful techniques, such as foundation models, it is essential to clarify not only their benefits but also their limitations and risks.

***So what is in the bottle?*** While CBMs should not be viewed as a universal solution to interpretability, they are a promising pathway toward the broader goal of interpretable and controllable machine learning architectures. This survey and the proposed taxonomy highlight a diverse and rapidly evolving body of work that reflects continuous progress across multiple dimensions of the CBM framework. By explicitly integrating human-interpretable concepts into the model's reasoning process, CBMs offer broad adaptability across application domains, but their effectiveness relies on specific assumptions about concept quality, completeness, and alignment. Looking ahead, advancing CBMs will require further coordinated innovation across multiple components, together with continued integration of ideas from related fields, to develop interpretable-by-design yet performant models suitable for broad real-world adoption. Importantly, although our work has identified various open challenges, the community has already demonstrated substantial progress in several of these areas, underscoring the strength, momentum, and potential of the CBM framework.

### Acknowledgments

This work was supported by the "ML2MT" project from the Volkswagen Stiftung, by the German Research Foundation (DFG) under Germany's Excellence Strategy (EXC 3066/1 "The Adaptive Mind", Project No. 533717223; and GRK 2853 "Neuroexplicit Models of Language, Vision, and Action", Project No. 471607914), the German Federal Ministry for Economic Affairs and Climate Action of Germany (BMWK), and by the German Federal Ministry for Research, Technology, and Space (BMFTR). It has further benefited from the HMWK projects "The Third Wave of Artificial Intelligence - 3AI", and Hessian.AI, the Hessian research

priority program LOEWE within the project WhiteBox, the EU-funded "TANGO" project (EU Horizon 2023, GA No 57100431), and from the Cluster of Excellence "Reasonable AI" funded by the DFG under Germany's Excellence Strategy EXC-3057.

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
