# OpenReview forum: "What’s in the Bottle? A Survey and Roadmap of Concept Bottleneck Models"
_TMLR — Accepted by TMLR_

### Review · Reviewer_e7ey · 2026-03-13

**Summary Of Contributions:**

This paper surveys the literature on Concept Bottleneck Models (CBMs) and proposes a taxonomy for categorizing the different components of CBM architectures. The paper separates a CBM into several modules, including the input, concept, and output modules. The input module includes elements such as the encoder and input embeddings. The concept module contains the concept representation itself and the mechanism used to produce concept activations, distinguishing between concept semantics (the definition and selection of the concept set) and concept grounding (how concepts are detected in the input data). Finally, the output module corresponds to the predictor that maps concept activations to the final task output, which is typically an interpretable model such as a linear classifier. Using this taxonomy, the authors categorize existing CBM approaches and discuss current challenges and open research directions.

One strength of the paper is its clear focus on CBMs as a distinct research area. While prior surveys often discuss CBMs within the broader concept-based explainable AI literature, this paper provides a more focused and extensive overview of CBM methods. The proposed taxonomy helps organize a growing body of work and provides a structured way to compare different CBM architectures. The paper also provides clear explanations of the CBM framework, its current challenges, and potential directions for future research. However, while many CBM approaches are referenced, a significant portion of the survey reads more like a listing of methods rather than a deeper discussion or analysis of their differences. In addition, the paper does not include empirical comparisons or benchmark results that would provide a clearer picture of the current state of the art in CBMs or how CBM-based approaches compare to standard non-CBM models. Including such comparisons could help readers better understand the practical performance trade-offs of these methods.

**Audience:**

Yes

**Audience Explanation:**

For readers unfamiliar with Concept Bottleneck Models (CBMs), this paper provides a clear introduction to the existing literature and the design space of CBM architectures. It gives readers an overview of current applications of CBMs as well as the key challenges and open research questions in the area. Given the recent growth of research on CBMs and concept-based interpretability methods, this survey could serve as a useful reference for researchers interested in understanding the current landscape and future directions of this line of work.

**Broader Impact Concerns:**

I do not see any specific ethical concerns requiring a broader impact statement, as the work is a survey of existing CBM literature.

**Claims And Evidence:**

Yes

**Claims Explanation:**

The primary claim of the paper is the introduction of a taxonomy for organizing the design space of Concept Bottleneck Models (CBMs) and providing a structured overview of the existing literature. The authors support this claim by reviewing a large number of CBM-related works and categorizing them according to their proposed architectural modules, such as the input, concept, and output components. The paper explains the distinctions within these modules, including the separation between concept semantics and concept grounding, and demonstrates how existing methods fit into the proposed taxonomy.

**Requested Changes:**

For the various tasks mentioned in the survey, it would be helpful to include a table summarizing the performance of representative CBM approaches. This could provide readers with a clearer sense of how different methods compare across tasks and datasets. If available, interpretability-related metrics could also be included alongside predictive performance to give a more complete view of the trade-offs involved.

While many CBM approaches are referenced throughout the survey, several are only briefly described. Expanding the discussion of some representative methods and highlighting their key methodological differences would improve the paper’s depth and help readers better understand the design choices within the CBM literature.

These changes are not strictly necessary for acceptance but would improve the usefulness of the survey as a reference for researchers.

---

> ### Author Response · Authors · 2026-04-15
> **Reponse to Reviewer e7ey**
>
> We thank the reviewer for their thorough evaluation and positive assessment of our paper, particularly for recognizing (i) the value of the proposed modular taxonomy for Concept Bottleneck Models (CBMs), (ii) the paper’s clarity and focused scope on CBMs as a distinct research direction, and (iii) the usefulness of our structured overview for organizing the growing body of CBM literature. We also appreciate the constructive suggestions aimed at further enhancing the paper’s depth and practical utility.
>
> ---
>
> ### 1. Performance and interpretability overview
>
> We agree that a consolidated table summarizing predictive performance and interpretability aspects could be informative. We initially explored this idea; however, compiling such a table in a rigorous and meaningful way proved infeasible due to the heterogeneity of the CBM literature.
>
> Specifically, existing CBM works vary widely across:
>  - domains and datasets (vision, language, tabular, multimodal),
>  - model backbones and training regimes,
>  - concept set definitions and sizes, and
>  - evaluation protocols, including differing notions of “concept accuracy” and interpretability metrics.
>
> These disparities make direct comparison unreliable and risk misleading readers. A fair comparison would require re-implementing or re-evaluating a substantial portion of prior work under standardized conditions, an endeavor beyond the scope of a survey paper. We note that such a benchmarking effort could itself constitute a valuable contribution for future work, once a clear and consistent evaluation framework emerges.
>
> To address this suggestion, we will explicitly clarify in the revised version that (a) this paper’s focus is conceptual synthesis rather than empirical benchmarking, (b) standardized interpretability and performance evaluation protocols for CBMs are still lacking, and (c) task- and domain-specific benchmarks would be required to make quantitative comparisons meaningful. This clarification should help set expectations and contextualize the absence of aggregated performance tables.
>
> ### 2. Expanded discussion of representative methods
>
> We appreciate the reviewer’s point that expanding the discussion of representative CBM approaches would provide readers with a deeper understanding of methodological differences and design trade-offs. Earlier drafts of the manuscript indeed contained more extensive method-level analysis, but these were condensed to maintain readability and emphasize the structural taxonomy.
>
> If wanted, we are glad to expand selected sections to restore some of this depth with a trade-off of readability. In particular, we would elaborate on key representative methods spanning the main categories of our taxonomy, for example, approaches emphasizing explicit concept supervision vs. those relying on implicit concept discovery, as well as works integrating generative or disentangled representations. While it is somewhat subjective which methods to highlight, we would prioritize influential and diverse ones to improve the survey’s explanatory balance.
>
> ---
>
> We thank the reviewer again for the constructive feedback and for acknowledging that these suggestions, while not essential for acceptance, would enhance the paper’s usefulness as a reference. We will incorporate the proposed clarifications to further strengthen the revised version.
>
> Best
> authors

---

### Review · Reviewer_p1u2 · 2026-03-14

**Summary Of Contributions:**

This paper presents a survey and roadmap of Concept Bottleneck Models (CBMs). The paper argues that the recent CBM literature has become fragmented, with methods differing along multiple axes such as concept acquisition, grounding, prediction, and training, and that the field lacks a sufficiently fine-grained unifying taxonomy. To address this, the authors review more than 100 CBM-related works and organize them into a modular taxonomy with four main modules: input, concept, output, and training. A central claimed novelty is the explicit separation of the concept module into concept semantics (which concepts are used and what they mean) and concept grounding (how concepts are connected to perceptual patterns in the data). The paper further surveys architectural choices, summarizes open challenges such as concept leakage, concept absence, intervention reliability, and robustness, and outlines future directions for CBMs.

**Audience:**

Yes

**Audience Explanation:**

The CBM literature has indeed expanded quickly, and a structured survey would be valuable for both newcomers and active researchers. The

**Broader Impact Concerns:**

I do not have major ethical concerns that would by themselves block publication. This is a survey on Concept Bottleneck Models, so the main issues are indirect rather than arising from a single deployed system.

**Claims And Evidence:**

Yes

**Claims Explanation:**

The paper is readable, well organized, and the proposed modular view is helpful. In particular, the distinction between semantics and grounding is a meaningful lens for discussing how concepts are defined versus how they are operationalized in data. The taxonomy figure and the large categorization table are useful artifacts.

**Requested Changes:**

1. Can the authors provide a formal review protocol? In particular, how were papers identified, screened, and included, and what were the explicit boundary conditions for calling a model a CBM or CBM-related work?

2. How reproducible is the categorization in Table 1? Was there any annotation rubric or inter-annotator agreement process?

3. Can the authors add one subsection that explicitly compares their taxonomy against the closest prior survey frameworks, using a few example papers to show where earlier categorizations become ambiguous and why the proposed taxonomy resolves that ambiguity?

4. Can the authors extract a few higher-level empirical observations from Table 1, rather than only using it as a catalog? Even a short “meta-findings” subsection would help.

5. Since the paper emphasizes interpretability as the defining CBM objective, can the authors more explicitly separate different interpretability criteria and discuss which surveyed methods support which notion?

---

> ### Author Response · Authors · 2026-04-15
> **Response to Reviewer p1u2**
>
> We thank the reviewer for highlighting the readability of the paper and the usefulness of the modular view, particularly the distinction between concept semantics and grounding.
> Below we address the requested points:
>
> ---
>
> ### 1. & 2. Review protocol and inclusion criteria.
> The survey was conducted through a structured but qualitative review process involving joint screening, analysis, and discussion of the literature. Papers were identified via keyword-based searches (e.g., “concept bottleneck model”) and citation chaining from seminal works such as Koh et al. [2020], as well as from closely related studies in the concept-based interpretability domain.
>
> Following the general guidance of Brocke et al. [1], we maintained a shared concept matrix where screened papers were logged with key attributes (e.g., modality, concept definition, grounding mechanism). The taxonomy evolved iteratively from this analysis and simultaneously defined the inclusion boundary: a work was considered CBM-related if it employed an explicit intermediate concept representation for prediction. Borderline cases were discussed to ensure consistency.
>
> While we did not quantify inter-annotator agreement, disagreements were resolved through discussion, effectively serving as a soft consensus-based process. The categorization was then rechecked during manuscript preparation. We note that our goal was not exhaustive coverage but rather a representative and structured mapping of the CBM design space.
>
> ### 3. Comparison to prior surveys:
> The taxonomy introduced in this work explicitly covers all elements of the CBM architecture, including the input, concept and output modules as well as the training procedure. The previous work of Mpinda et al. [2026] exclusively distinguishes single- and multilabel tasks and disregards all components of the CBM architecture. The survey of Wang et al. [2026] categorizes models by a mix of the concept module and evaluation criteria, ignoring all other architectural elements of a CBM. Besides that, our taxonomy provides a more detailed insight into the concept module by separating concept semantics and grounding.
>
> This enables a more detailed and extensive comparison of CBM methods. All differences in input and output module would not appear in the other categorizations. Regarding the concept module, our taxonomy allows, for example, to understand the differences between the concept source of Chowdhury et al. [2024] and Oikarinen et al. [2023], where the former uses a VLM for semantics and for grounding, while the latter uses an LLM for semantics and a VLM for grounding. In previous surveys, these differences are not visible, as they are both labeled with the same concept source in the categorization by Wang et al. [2026]. Generally, this distinction is quite relevant as roughly ~40% of the papers in our list use different approaches for semantics and grounding.
>
> We have expanded on this differences in Sec. 3 to make the differences between our introduced taxonomy and the previous surveys in more detail.
>
> ### 4. Meta Findings
>
> We agree that extracting higher-level observations strengthens the paper. To address this, we added a concise new section, “4.5 Summary of Observed Trends,” which distills the main patterns emerging from our taxonomy and serves as a concluding synthesis for Section 4.
>
> ### 5. Interpretability Criteria
> We thank the reviewer for this suggestion. We have now separated "Performance Evaluation and Benchmarking" and "Evaluating Interpretability" from the previous section “Evaluation and Benchmarking“ to highlight aspects of interpretability in CBM challenges.
> The fundamental challenge regarding this is the absence of a shared definition of interpretability itself. Different works operationalize interpretability differently, ranging from concept consistency and faithfulness to intervention effectiveness and sparsity, without consensus on what these mean or how they relate. We now discuss the problems this creates in the new section.
>
> ---
>
> We thank the reviewer for their very constructive feedback and hope our responses adequately address the raised points.
>
> Best
> authors
>
> [1] Brocke, Jan vom, et al. "Reconstructing the giant: On the importance of rigour in documenting the literature search process." (2009).

---

### Review · Reviewer_DYPr · 2026-04-08

**Summary Of Contributions:**

The authors survey the state of research on Concept Bottleneck Models, creating a taxonomy to compare different works. Then they introduce different axes of variation, and challenges related to different parts of the taxonomy. They also provide forward-looking ideas about open challenges and questions for the field.

**Audience:**

Yes

**Audience Explanation:**

I had a quick look at the current state of surveys on the field, and I concur with the authors that a survey of this scope and quality is missing. I personally would also share this paper as a starting survey to people new to the field.

In addition, the open challenges and connections to other fields are well-presented and thought through, in my opinion. This makes the survey possibly also interesting to people already longer in the field (I certainly found a few papers I wasn't aware of that are relevant to my research).

**Claims And Evidence:**

Yes

**Claims Explanation:**

This is, in my opinion, a thorough and well written survey, and I believe overall it's of high quality. The authors give clear opinions and a clear view of the field, without resorting to overenumeration, making for a pleasant read.

There are few parts I could find that were unconvincing or inaccurate, at least from what I know of the field. I will give some examples later, but they are minor enough to answer yes here.

**Requested Changes:**

I think the connection to two fields could be expanded on:
- Neurosymbolic AI. In particular, the subfield of neurosymbolic predictors is very closely aligned to concept-based models (they are CBMs with discrete concepts and a program as predictor, as mentioned in Section 6.2). See for instance [1, 2] on reasoning shortcuts, which connects to CBMs very explicitly. In that sense, neurosymbolic methods are an alternative measure to deal with finding concept annotations under "Assuming Concept Annotations", as background knowledge in the form of programs can provide extra weak supervision. These papers also directly connect to the symbol grounding problem discussed in this survey. Finally, the use of "reasoning shortcuts" (page 15) might be different from that meant in [1, 2], although Bortolotti 2025 discusses a variation of them ("joint" reasoning shortcuts). A paper that fits well in 'program-based reasoning' is LEFT [4].
- Causal representation learning. This field is not mentioned at all, but also aims to find disentangled representations. In particular, a recent paper [3] connects this explicitly to concept based models

Other points
- I'm curious why zero-shot generalisation is not a possible advantage of CBMs. Learning concepts could allow for easy transfer to new classes if concepts are properly trained.
- Section 2, central challenges: I was missing concept leakage here, but it was discussed later. Curious why not also here.
- Section 4.3.1: Where are concept embedding models in this taxonomy? Or do you not consider them in this survey? If not, I think they deserve a mention to prevent confusion
- Section 5.1 (Concept Absence): I think this could be connected better to the field of epistemic AI, which studies eg decompositions between aleatoric and epistemic uncertainty
- Section 5.1 (Grounding, Concept L...): "Bottleneck fails to enforce conditional independence": I think this is inaccurate, it's not so much that the conditional independence is violated (which is true by design), but rather that there is a lot of mutual information between c and x (or c and y), at least more than we want. See eg Almudevar 2026 et al (ICLR), which is already cited
- Section 6.1, Expanding CBMs: The problem of representing eg temporality, modality and hierarchy is studied in the field of knowledge representation (from symbolic AI), this could be mentioned for future inspiration.
- Section 6.1, "Dynamic CBMs": The connection to system 1 and system 2 (which is already a controversial theory) seems a bit flimsy to me. Why would CBMs be more 'system 2'?
- Section 6.2, first paragraph: "differentiable" -> "differentiated"?
- Section 6.2, connection to SAEs: The ActivationReasoning paper could be mentioned here as it uses SAEs to identify concepts. Also [5] could be mentioned which explicitly connects CBMs to SAEs formally.
- Conclusion: It would be fun if the referrence to "What's in the bottle" is also in the introduction :-)

Some missing literature that's relevant:
 - Hsu, Joy, et al. "What Makes a Maze Look Like a Maze?." International Conference on Learning Representations (ICLR), 2025.
- Mao, Jiayuan, Joshua B. Tenenbaum, and Jiajun Wu. "Building Intelligent Agents with Neuro-Symbolic Concepts." Communications of the ACM 69.2 (2026): 69-79.
- Martínez-García, María, et al. "A Probabilistic Hard Concept Bottleneck for Steerable Generative Models." The Fourteenth International Conference on Learning Representations.

1. Marconato, Emanuele, et al. "Not all neuro-symbolic concepts are created equal: Analysis and mitigation of reasoning shortcuts." Advances in Neural Information Processing Systems 36 (2023): 72507-72539.
2. Marconato, Emanuele, et al. "Symbol grounding in neuro-symbolic ai: A gentle introduction to reasoning shortcuts." arXiv preprint arXiv:2510.14538 (2025).
3. Fokkema, Hidde, Tim van Erven, and Sara Magliacane. "Sample-efficient learning of concepts with theoretical guarantees: from data to concepts without interventions." NeurIPS (2025).
4. Hsu, Joy, et al. "What’s left? concept grounding with logic-enhanced foundation models." Advances in Neural Information Processing Systems 36 (2023): 38798-38814.
5. Fel, Thomas, and Gianni Franchi. "A Geometric Unification of Concept Learning with Concept Cones." arXiv preprint arXiv:2512.07355 (2025).

---

> ### Author Response · Authors · 2026-04-15
> **Response to Reviewer DYPr**
>
> We thank the reviewer for their thorough and positive assessment — in particular for noting the survey's readability, the clarity of our taxonomy, and the value of the open challenges section. We are glad the paper could surface relevant works even for readers already active in the field.
>
> Below we address each of your points in turn. All corresponding changes are also visible in the updated manuscript uploaded alongside this rebuttal.
>
> ---
>
> ### Connection to Neurosymblic AI
>
> We thank the reviewer for these pointers and have integrated the suggested works across the paper (e.g., sections 5.1, 5.2, 6.1, and 6.2). We have intentionally kept the neurosymbolic discussion focused rather than comprehensive: a thorough treatment of neurosymbolic AI would warrant its own survey, and our aim here is to position CBMs in relation to it without expanding the scope of the paper.
>
> ### Connection to Causal Represenation Learning
> We have added a dedicated paragraph on causal representation learning to Section 6.2, discussing how it relates to concept-based models.
>
>
> ### Minor Changes
>
> - **Zero Shot Generalization:** We agree this is a valid point and have added a discussion of this to the generalist/specialist section in the revision. We note, however, that this falls short of true zero-shot generalization as achieved by models like CLIP, since the predictor still requires supervision for new classes.
> - **Concept Leakage:** The goal of section 2 is to give a general overview of CBMs, their main advantages and the main challenges when applying them. While leakage is an important challenge in particular for jointly trained CBMs, it is also a bit more "unexpected" and does not fit the more general introduction of CBMs in section 2.
> - **Concept Embedding Models:** We do consider CEMs as a subgroup of CBMs and have referenced several of these works within the scope of this survey. We argue that all CEM parts can be mapped to the taxonomy components, as the concept embeddings are primarily used to obtain presence and absence "activations" for each concept, and their supervised training procedure falls under concept grounding.
> - **Epistemic AI:** We have accordingly made a connection in section 5.1 to epistemic uncertainty.
> - **Conditional independence:** There is a direct connection between the conditional independence of x and y given c $(y \perp x \mid c)$ and the conditional mutual information between x, y and c: $I(x; y \mid c) = 0$, i.e. the conditional mutual information is zero is equivalent to the conditional independence [Havasi et al. 2022]. And while this independence should ideally be given, in practice it is usually not the case, for example when the given concept set does not encode all information necessary to successfully predict $y$. In exactly these cases, a jointly trained CBM is prone to leakage, as the joint optimization encourages to encode the remaining information into the concept representation. High mutual information between $x$ and $c$ does not generally mean leakage: In the extreme case that $x$ and $c$ are the same, the mutual information between them is 1, but this does not mean that leakage occurs.
> However, we agree that the work of Almudévar et al. [2026] should be mentioned in this context and included it.
> - **Connection to Knowledge Representation:** Thanks for pointing this out, we mention this connection in the section.
> - **Connection to System 1 & 2:** In this discussion point, we do not argue that CBMs are more "system 2" per se, but rather that a potential direction for future research can move into the direction of "On-Demand CBMs". This would mean that the default computation mode for a model is an opaque setting, but there is an option to include an explicit concept bottleneck if necessary, which then allows for more controlled and detailed reasoning and inspection. In such a scenario, including the CBM computation mode would correspond more to the slow and controlled "system 2" mode.
> - **Section 6.2**: Thanks for pointing these things out, we updated the section accordingly.
> ---
> We thank the reviewer for their constructive feedback and hope our responses adequately address the raised points.
>
> Best
> authors

---

### Decision · Action_Editor_RKph · 2026-05-19

**Recommendation:** Accept as is

**Audience:**

Yes

**Audience Explanation:**

The reviewers agree that the field of CBMs is lacking a survey of this scope and quality and welcome the contribution.

**Claims And Evidence:**

Yes

**Claims Explanation:**

The reviewers found the paper claims to be supported and to present clear evidence.

The paper presents a survey with clear opinions and overview of Concept Bottleneck Models (CBMs). It is well organized and the modular view is helpful. Moreover, the organization of the body of work on CBMs and the proposed taxonomy is helpful as well.